# The bound growth of induced earthquakes could de-risk hydraulic fracturing
Ryan Schultz [1] ✉, Federica Lanza[1], Ben Dyer[2], Dimitrios Karvounis [2], Rémi Fiori[2], Peidong Shi [1], Vanille Ritz [1], Linus Villiger [1], Peter Meier[2] & Stefan Wiemer [1]

The world's energy supply depends critically on hydraulic fracturing (HF) to access otherwise uneconomical resources. Unfortunately, HF also has the potential to induce larger earthquakes – with some projects being prematurely terminated because of perceived earthquake risks. To de-risk HF, we use a suite of statistical tests to discern if some physical process has restricted the growth of earthquake magnitudes. We show that all stage stimulations at both UK PNR-1z and Helsinki St1 indicate bound fracture growth, implying a more controllable operation. Contrastingly, stimulations at Utah FORGE and UK PNR-2 sequentially transitioned into unbound fault reactivation. The problematic stages (that ultimately led to the termination of PNR-2) are clearly distinguishable. We postulate that our research can discriminate fracture stimulation from fault reactivation, contributing to the de-risking of HF operations worldwide. Our statistical tests provide a framework for model falsification, which can guide physical insights into the bounding processes.

The insatiable demand for energy continues to grow each year. In recent decades, hydraulic fracturing (HF) has been revolutionary for unlocking trapped hydrocarbon energy sources: fluids injected under high-pressure cause subsurface rocks to fail, stimulating fractures that act as permeable fluid pathways needed to extract resources[1]. This stimulation process can either propagate failures in tension or in shear; these failures can happen on either pre-existing fractures or newly created fractures. However, the urgency for more energy is also complicated by climate goals and decarbonisation needs. Enhanced Geothermal Systems (EGS)[2] could play a part in the energy transition, since they have the potential to fill this niche by producing abundant, green, sustainable, baseload power[3]. EGS taps the limitless geothermal potential of the deep earth through stimulation techniques (e.g., HF)[4]: stimulated fractures can also act as fluid pathways needed for heat exchange. EGS could be on the cusp of widespread adoption, considering recent successes[5]. However, the deployment of HF techniques is not without risks.

The HF process also has the potential to induce earthquakes[6,7]. If stimulated fractures intersect a pre-existing fault, then the increase of pore pressure can cause fault slip reactivation[8]. These earthquakes can be consequential via economic/human losses, premature project termination, or resource development moratoriums[9,10]. For example, magnitude ~5 earthquakes have caused hundreds of millions of dollars in economic losses from both petroleum and EGS operations using HF[10,11]. On the other hand, there are also numerous HF projects that did not encounter detrimental earthquake impacts[12–15]. Thus, there is a clear need to effectively manage these earthquakes and mitigate their consequences.

Guidelines to managing induced earthquake risks exist[16,17]; these provide templates for pre-screening assessments, risk evaluation, monitoring, and defining operational endpoints. The core risk management framework is the traffic light protocol, which details how an operation should proceed, depending on the largest magnitude observed[18]. Risk-based approaches can identify magnitude thresholds for when an operation must stop, to remain below acceptable risk tolerances[19–21]. Despite these efforts, state-of-the-art procedures can still fail to robustly discern and then react to risky/consequential projects[16,17] – as evidenced by high-profile cases that caused nuisance, damage, human losses, premature project termination, or development moratoriums[9–11].

Fundamental to this risk management process is accurately anticipating the growth of the largest earthquakes[22]. Unfortunately, this is a poorly understood topic in seismology. What is generally accepted, is that the statistics of earthquake magnitudes (M) typically follows the Gutenberg-Richter magnitude frequency distribution (GR-MFD)[23]. A natural consequence of the GR-MFD is that the largest magnitude events ($M_{LRG}$) scale with the size of the catalogue ($N$)[24]. It is debatable whether any other physical process bounds the growth of magnitudes: i.e., an upper bound magnitude ($M_{MAX}$) that restricts the (statistically expected) growth of magnitudes. Some $M_{MAX}$ models have been hypothesised. In the simplest model, an $M_{MAX}$ can arise from the finite size of the fault extent, since magnitude is tied to shear slip across a fault area[25]. More contentiously, $M_{MAX}$ could also be some injection-dependent process that changes with time; for example, either evolving with seismic moment release (McGarr-like models)[26,27] or scaling with self-arrested rupture dynamics (Galis model)[28]. Investigating

[1]Swiss Seismological Service, ETH Zürich, Zürich, Switzerland. [2]Geo-Energie Suisse, Zürich, Switzerland. ✉e-mail: Ryan.Schultz@sed.ethz.ch

the physics of these $M_{MAX}$ models would benefit from approaches that could verify/falsify their presence in a catalogue.

However, there is a lack of empirical approaches to identify/falsify these $M_{MAX}$ models – the identification of $M_{MAX}$ solely from a catalogue is fraught with difficulty. Traditional approaches to this $M_{MAX}$ problem have found it to be solvable only in ideal conditions[29,30], often requiring a significant degree of truncation for robust identification. On the other hand, verifying the existence of $M_{MAX}$ within an induced earthquake sequence is significant, because then the risk of induced seismicity is inherently more controllable[22]. Given this gap between need/difficulty, expert opinion panels on $M_{MAX}$[31] often substitute for quantitative science out of necessity.

In this study, we consider a reformulation of this $M_{MAX}$ problem. Here, we implement a novel suite of statistical tests to discern if a sequence of earthquake magnitudes was bound by an $M_{MAX}$[32]. These tests compare event magnitudes (M) against 'jumps' in the sequence of largest earthquakes ($\Delta M_{LRG}$), to infer the presence of a maximum upper bound ($M_{MAX}$). Simply put, the statistical distributions of M and $\Delta M_{LRG}$ are identical when a sequence is unbound, but distinguishable in the presence of an $M_{MAX}$ (Fig. 1)—this is because the presence of an $M_{MAX}$ restricts the allowable sizes of $\Delta M_{LRG}$. Our suite of three statistical tests exploits this fact. The first test uses the Kolmogorov-Smirnov test (KS-test) to compare M and $\Delta M_{LRG}$ observations, to discern if they are drawn from the same distribution; this tests for the existence of an $M_{MAX}$. The second test uses a maximum likelihood estimation (MLE-test) to quantify the value of $M_{MAX}$. The third test computes ensemble weights (EW-test) through Akaike and Bayesian Information Criteria; this selects the best $M_{MAX}$ model amongst several hypothesised choices, given the catalogue data. We refer to this suite of three novel tests as CAP-tests. These CAP-tests are generally more sensitive to $M_{MAX}$ bounds than traditional approaches[32]. Full details on terminology, methods, guidelines to the interpretation of results, and synthetic examples are all available in the Supplementary Information (Text S1–S4).

We demonstrate how this workflow (Fig. 2 & S1) can be used to identify concerning cases of induced seismicity. This is accomplished by contrasting cases of induced seismicity related to HF (Fig. S7). In the first case, we consider Preston New Road 1z (PNR-1z)[33]: a shale gas HF operation in the UK that injected ~4200 m³ over 17 (of 41 planned) stages and produced earthquakes up to $M_L$ 1.6 in 2018. In the second case, we consider Helsinki St1[15]: an EGS operation in Finland that injected ~18,200 m³ over five stages and produced earthquakes up to $M_W$ 1.9 in 2018. Our tests find that all stage stimulations in these two cases follow a bound process, dominantly suggesting fracture growth via stage stimulations. In the contrasting cases, we consider data from the Frontier Observatory for Research in Geothermal Energy (FORGE)[34] and Preston New Road 2 (PNR-2)[35]. FORGE is an EGS operation in Utah that injected ~18,500 m³ over ten stages and produced earthquakes up to $M_W$ 1.8 in 2022 and 2024. PNR-2 is a shale gas HF operation in the UK that injected ~2500 m³ over seven stages and produced earthquakes up to $M_L$ 2.9 in 2019. Our tests find that the first handful of stages follow a bound process (i.e., restricted by a volume-based $M_{MAX}$). However, the subsequent (and problematic) stages were clearly unbound, dominantly suggesting fault reactivation via stage stimulations. The PNR-2 case was prematurely terminated because of induced seismicity concerns.

## Bound growth
### Bound growth at PNR-1z

HF was used in the UK at three wells, to extract the hydrocarbons locked within the Mississippian aged Bowland Shale. The operation at PNR-1z, near Blackpool, was the second HF operation in the UK[33]. The first set of stimulations (October–November 2018) produced moderate seismic response; these events were concerning due to their magnitude above the red-light threshold, which caused an operational pause. The second set of stimulations (December 2018) resumed after this pause. The largest event ($M_L$ 1.6) occurred on 11 December 2018[33]. All events appear to be

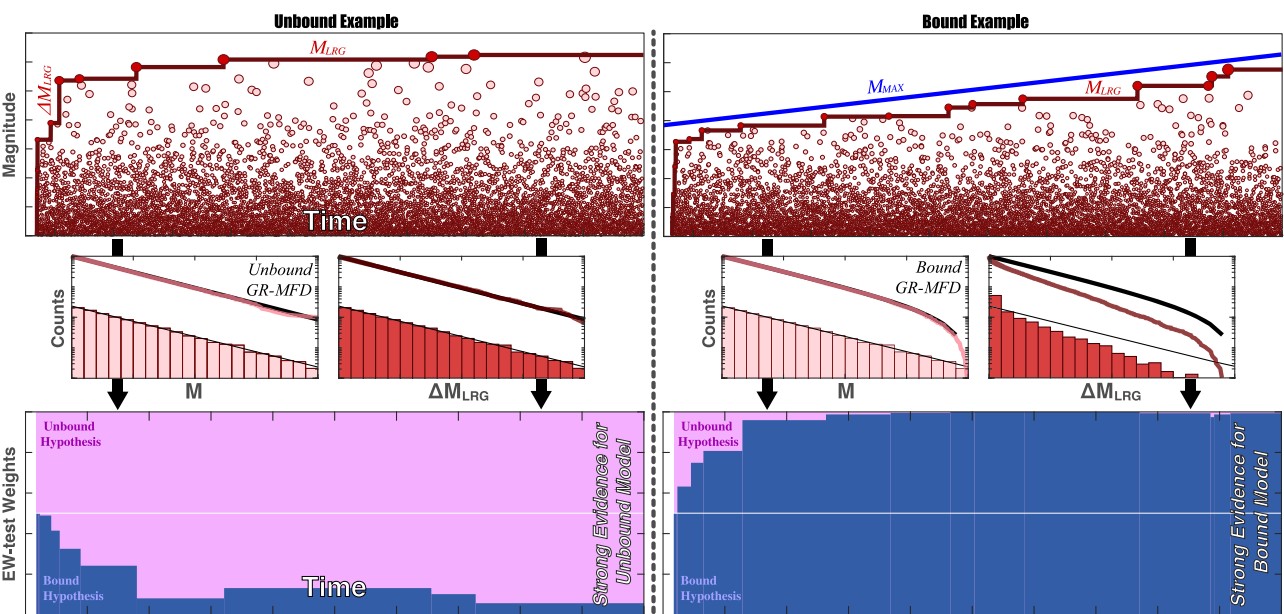

**Fig. 1 | Schematic demonstration of observational and methodological concepts.** The upper panels show hypothetical catalogues of earthquakes. Earthquake magnitudes (M, pink circles), the sequence of largest events ($M_{LRG}$, red line), and jumps in the largest event magnitudes ($\Delta M_{LRG}$) can be directly observed. Potentially, some unobservable physical process could be bounding these catalogues ($M_{MAX}$, blue line). Left and right panels detail contrasting concepts for an unbound and bound catalogue, respectively. If there is an $M_{MAX}$, then it can be statistically inferred from the observables. Middle panels show the differences between distributions of M and $M_{LRG}$, when bounded by $M_{MAX}$ (or not). The distribution of M follows a GR-MFD: analytical cumulative (thick black line) and non-cumulative (thin black line) agree

with numerical cumulative (pink line) and non-cumulative (pink bars) distributions. The distribution of $\Delta M_{LRG}$ will differ from the GR-MFD when bounded by $M_{MAX}$. Bottom panels show the results of EW-tests using these concepts. Weights of two hypothesised models, an unbound $M_{MAX}$ (pink area) and a bound $M_{MAX}$ (blue area), change as new values of $M_{LRG}$ are observed. EW-tests can quickly infer the presence of the true $M_{MAX}$ model, from the equivocal a priori assumption (white horizontal line). Evidence for a model 'winning' over another becomes substantial/ positive, strong, and decisive when the ratio-of-weights (i.e., odds ratio) is 3, 10, and 100, respectively.

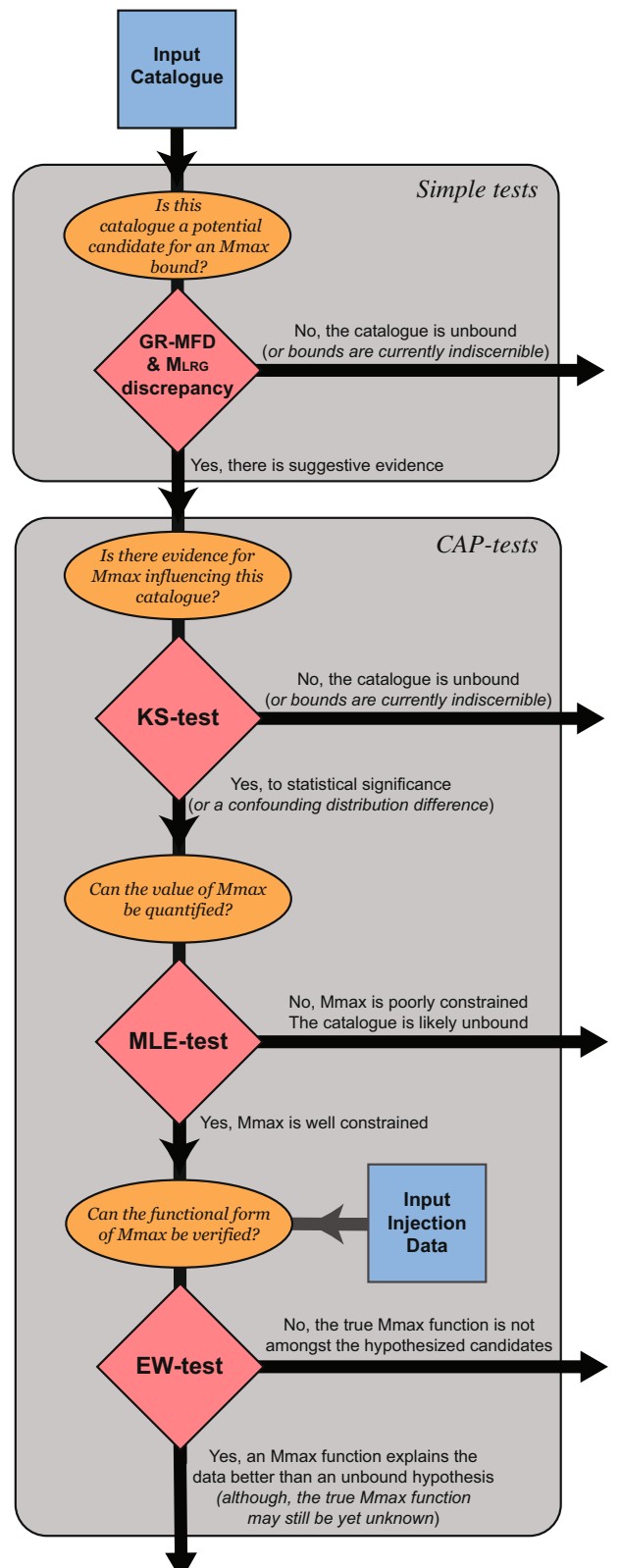

**Fig. 2 | Logical sequence of statistical tests to discern $M_{MAX}$ bounds.** Simple-tests are used to pre-screen for potentially bound catalogues by considering the degree-of-truncation. Afterwards, CAP-tests can more rigorously identify bound cases.

hydraulically linked into a single cluster (Fig. S8; Text S5.1). The success of operations at PNR-1z enabled the subsequent operation at PNR-2.

To begin assessing if some process might be restricting magnitude growth, we fit the GR-MFD[23] and perform the simple-tests (Text S5.1). The

earthquakes here appear to be deficient in large magnitude events (Fig. S9). Correspondingly, the discrepancy between the largest observed event and the largest expected event magnitudes is $-0.5\,M_W$, which is a rare occurrence ( ~ 2%) purely by chance. These initial assessments are suggestive of some $M_{MAX}$ upper bound restricting the PNR-1z catalogue growth.

Next, we use the CAP-tests to detect and assess the potential for $M_{MAX}$ more rigorously (Fig. 3 & S10). Average KS-test confidences are 99.82% (Text S5.1), where 95% is a common threshold to declare statistical significance. Next, we seek test hypotheses for the functional form of $M_{MAX}$: our EW-tests focus on a McGarr-like model (i.e., log-proportional to injected volume; $M_{MAX} \propto \log_{10}(V^1)$)[26,27], the Galis-like model (i.e., $M_{MAX} \propto \log_{10}(V^{3/2})$)[28], and the unbound null hypothesis. Similarly, EW-tests also show strong evidence for an $M_{MAX}$ bound process. By ~100 event observations, respective model weights are already {0.39, 95.69, 3.92%} which means the Galis-like $M_{MAX}$ model is ~24 times more likely than unbound ($\log_{10}$ odds ratio of $+1.4$; Fig. 3a). By the end of stimulations, the Galis-like model is >100 times more likely than the unbound model (Fig. 3a) —which surpasses the definitive evidence threshold.

**Bound growth at Helsinki St1**. HF was used in a deep geothermal well to create an EGS in the Precambrian Svecofennian basement rocks near Helsinki, Finland. The St1 operation injected ~18,200 m³ of fluid within five stages from June-July 2018[15]. These stimulations produced a moderate seismic response; the largest event ($M_W$ 1.9) occurred on 8 July 2018. All events appear to be hydraulically linked into a single cluster (Fig. S11; Text S5.2), which is consistent with prior interpretations[15]. The success of this operation allowed for later restimulation of the reservoir.

To assess if some process might be restricting magnitude growth, we first employ the simple tests based on GR-MFD analyses (Text S5.2). The earthquakes here appear to be deficient in large magnitude events (Fig. S12). Correspondingly, the magnitude discrepancy between the largest observed event and the largest expected event is -0.7 $M_W$, which is a rare occurrence ( < 1%) purely by chance. These initial assessments are suggestive of some $M_{MAX}$ upper bound restricting the St1 catalogue growth.

Next, we use the CAP-tests to detect and assess the potential for $M_{MAX}$ more rigorously (Fig. 4 & S13). Average KS-test confidences are 98.17% (Text S5.2). Similarly, EW-tests also show strong evidence for an $M_{MAX}$ bound process (Fig. 4): McGarr-like and Galis-like models are >100 ($\log_{10}$ odds ratio of $+2.2$) and ~57 ($\log_{10}$ odds ratio of $+1.8$) times more likely than unbound models, respectively. This dataset is somewhat ambiguous between either McGarr-like or Galis-like models best explaining the data.

We note that we have examined supporting cases, including the EGS stimulations at Soultz-sous-Forêts in France[36]. There, we also find some evidence for a bound process restricting the growth of earthquake magnitudes during their stimulations (Text S5.5; Figs. S27–S36).

## The transition to unbound growth
### The transition to unbound growth at FORGE
Utah FORGE is a geothermal experiment ( ~ 16 km NE of Milford) aimed at commercialising EGS[34]. Two deviated wells form a doublet heat exchange system (16 A injector & 16B producer). The first three stages of well 16 A were stimulated in 2022; in 2024, stage 3 was restimulated, followed by the remaining stimulations (Figs. S14-S15). Originally, nine stages were planned for well 16 A at FORGE. However, frack barriers were encountered at stage 6, preventing fracture stimulation; in response, a tenth stage was added. Correspondingly, stage 6 separates the adjacent stages into two hydraulically separate compartments. Overall, stages are compartmentalised into hydraulically independent clusters as follows: cluster 1 (stages 1-2), cluster 2 (stages 3–6), and cluster 3 (stages 7–10). Additional justifications for this clustering choice are provided in the Supplementary Information (Text S5.3) and in prior studies[35].

To assess the potential for bound/unbound earthquake growth, we first employ the simple GR-MFD analyses (Text S5.3). The earthquakes here appear systematically skewed in larger magnitude events: both in terms of the full catalogue (Fig. S16) and on a cluster-by-cluster basis (Fig. S17).

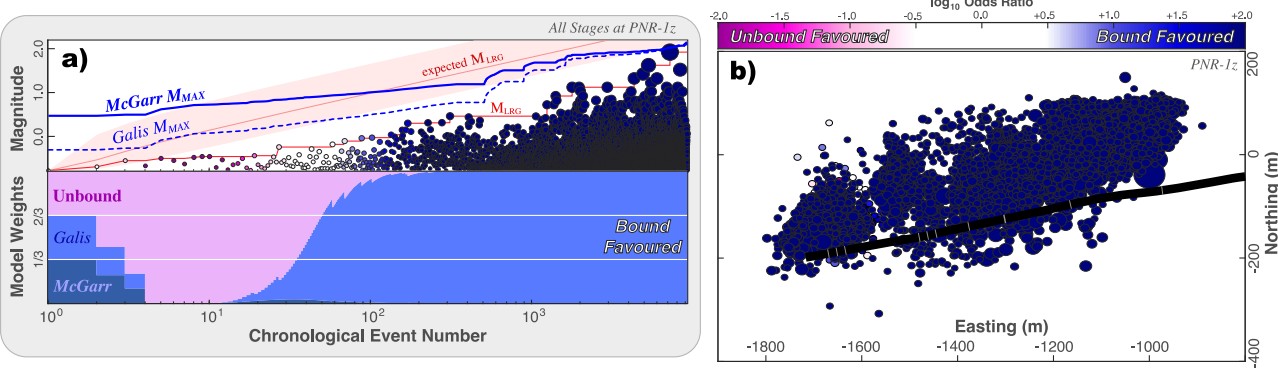

**Fig. 3 | EW-tests applied to PNR-1z. a** EW-tests are shown for all the stages at PNR-1z. The input data includes earthquake magnitudes (circles), the sequence of observed $M_{LRG}$ (red line), alongside the expected $M_{LRG}$ at the 10/50/90 percentiles (red area); two possible injection-based $M_{MAX}$ models are considered (blue lines). Correspondingly, the ensemble weights from the EW-test are shown; bound models are highly favourable. **b** Map view showing the locations of earthquakes (circles) alongside the well trajectory (black line) and stage locations (grey lines). Throughout the figure, all earthquakes are colour-coded accordingly to their likelihood of being bound or unbound from EW-tests.

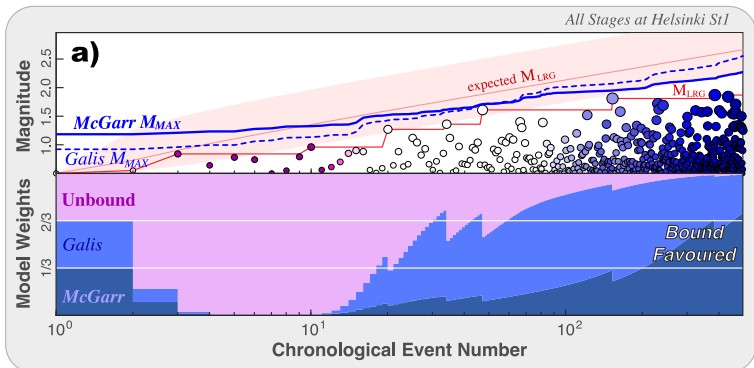

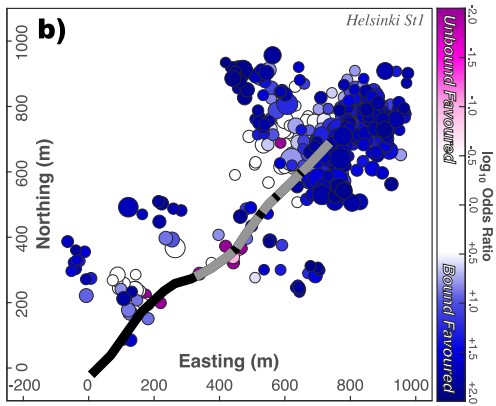

**Fig. 4 | EW-tests applied to Helsinki St1. a** EW-tests are shown for all the stages at Helsinki St1. The input data includes earthquake magnitudes (circles), the sequence of observed $M_{LRG}$ (red line), alongside the expected $M_{LRG}$ at the 10/50/90 percentiles (red area); two possible injection-based $M_{MAX}$ models are considered (blue lines). Correspondingly, the ensemble weights from the EW-test are shown; bound models are highly favourable. **b** Map view showing the locations of earthquakes (circles) alongside the well trajectory (black line) and stage locations (grey lines). Throughout the figure, all earthquakes are colour-coded accordingly to their likelihood of being bound or unbound from EW-tests.

Furthermore, we also consider the discrepancy between the largest observed event and largest expected event magnitudes. Correspondingly, this difference ranges between -0.4 and -1.3 $M_W$ for the first two clusters, but is +0.1 $M_W$ for the third cluster. These are rare occurrences ( ~ 1%) for the first two clusters. Together, these initial assessments are suggestive of some $M_{MAX}$ upper bound restricting the FORGE catalogue growth for clusters 1-2.

To more rigorously examine $M_{MAX}$ upper bounds, we use the suite of CAP-tests. Both KS-test and MLE-test results provide strong evidence for the existence of $M_{MAX}$ upper bounds. Specifically, KS-test confidences range between 99 and 100% for clusters 1 and 2. Thus, there is strong evidence for some physical $M_{MAX}$ process. On the other hand, cluster 3 does not meet the 95% confidence threshold. For the EW-tests on stage 3 in 2022, within ~100 event observations, respective model weights are already {70.44, 17.69, 11.87%} which means that the McGarr-like $M_{MAX}$ model is ~6 times more likely (log₁₀ odds ratio of +0.8; Fig. 5a). Furthermore, clusters 1 and 2 show strong evidence for a bound process after EW-tests (Fig. 5). Specifically, the McGarr-like model appears to best explain these bound clusters (Figs. S18–S21). On the other hand, the EW-tests for cluster 3 show strong evidence for an unbound process: the unbound model is ~50 times more likely than the best-fitting bound model. These results hold, even after various perturbation/sensitivity tests (Text S6–S7).

**The transition to unbound growth at PNR-2**

The operation at PNR-2, near Blackpool, was the most seismically problematic[37] and last of the three HF operations in the UK. The first 1–3 stages only produced moderate seismic response in the western-most clusters, while more concerning seismicity was encountered at the eastern-most clusters, which predominantly occurred during stages 5–7 (Fig. S22). The eastern-most cluster culminated in the $M_L$ 2.9 event on 26 August 2019, which occurred >72 hours after the shut-in of stage 7[37]. This event caused the premature termination of PNR-2, after only the first seven stages (of 47 planned) were completed. Ultimately, these earthquakes spurred a resource development moratorium in the UK[38].

To assess the potential for bound/unbound earthquake growth, we repeat the simple GR-MFD analyses. The western-most clusters appear to be deficient in large magnitude events, while the eastern-most cluster is abundant (Figs. S23-S24). Correspondingly, the discrepancy between the largest observed event and largest expected event magnitudes is -1.2 $M_W$ and +0.8 $M_W$, respectively. This is a rare occurrence ( < 1%) for the western-most cluster, but not the eastern-most. Together, these initial assessments

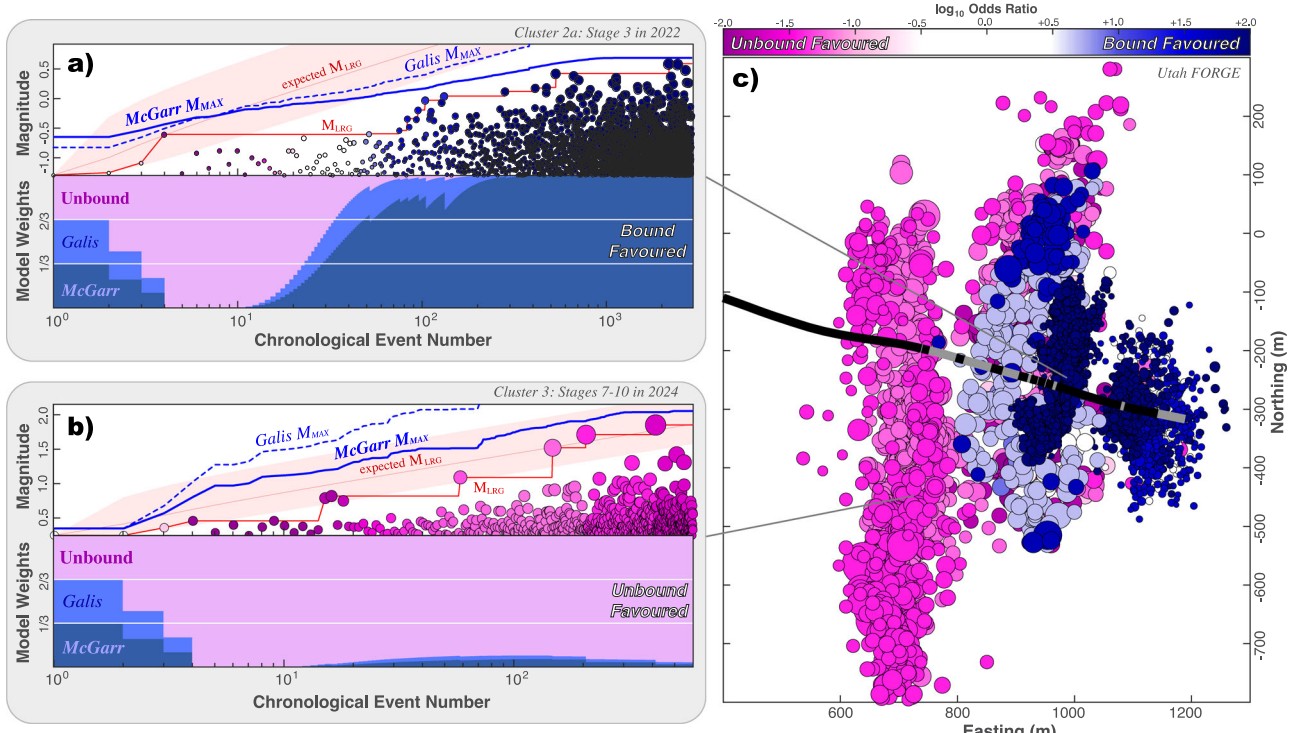

**Fig. 5 | EW-tests applied to FORGE.** EW-tests are shown for **a** the 2022 stage 3 and **b** for the 2024 cluster 3 (stages 7–10). The input data includes earthquake magnitudes (circles), the sequence of observed $M_{LRG}$ (red line), alongside the expected $M_{LRG}$ at the 10/50/90 percentiles (red area); two possible injection-based $M_{MAX}$ models are considered (blue lines). Correspondingly, the ensemble weights from the EW-test are shown; bound models are highly favourable in most cases (Figs. S18–S21). **c** Map view showing the locations of earthquakes (circles) alongside the 16 A well trajectory (black line) and stage locations (grey lines). Throughout the figure, all earthquakes are colour-coded accordingly to their likelihood of being bound or unbound from EW-tests.

are suggestive of some $M_{MAX}$ upper bound restricting the PNR-2 catalogue growth at the western-most cluster, but not the eastern-most.

Again, we use the CAP-tests to detect and assess the potential for $M_{MAX}$ more rigorously (Fig. 6). KS-test and MLE-test results show strong evidence for $M_{MAX}$ in the western-most case (i.e., > 99.99% confidence), but not the eastern-most case (Text S5.4). Similarly, EW-tests also show strong evidence for a bound process, but only for the western-most cluster (Fig. 6a). The western-most cluster is somewhat ambiguous between either McGarr-like or Galis-like models best explaining the data. On the other hand, the eastern-most cluster definitively indicates an unbound process, being ~100 times more likely than the best-fitting bound model.

### Implications

**Practical risk management.** Induced earthquakes pose considerable risks. However, the chances of encountering earthquake consequences for a bound/unbound stage stimulation are significantly different (Text S8; Fig. S41). Thus, the ability of CAP-tests to discern between these two extremes constitutes an important reactive risk mitigation measure: bound stages could proceed as planned, while unbound stages could be treated with caution (or even prematurely stopped and then skipped). This would be especially important for cases where earthquake magnitude growth is approaching the red-light[19–21]—the last-possible stopping-point before a regulatory intervention is required. In this sense, CAP-tests provide information on how/if 'jumps' in earthquake magnitudes (or trailing seismicity) are restricted, which is consistently identified as an important factor for managing induced seismicity[22,39,40].

To elaborate, we separate bound/unbound clusters using just the EW-test results (Fig. S42). We use an odds ratio threshold of 3-to-1 (i.e., $\log_{10}$ of +0.5) to discern a bound case to statistical significance; all subsequent $\Delta M_{LRG}$ events are labelled bound. Clusters below this threshold are considered unbound. We then examine the distribution of $\Delta M_{LRG}$ from these two catalogue subsets (Fig. 7); we find that they are different to statistical significance (KS-test confidence 99.8%). The unbound cases are well-fit to the GR-MFD, with a typical $b$-value of 1.73 ± 0.18 ($R^2$ 0.989) and $\Delta M_{LRG}$ magnitude jumps up to +1.2. On the other hand, the bound cases are not well-fit to the GR-MFD, with an 'unphysical' $b$-value of 3.02 ± 0.30 ($R^2$ 0.967) and $\Delta M_{LRG}$ abruptly truncated near +0.3.

First, this finding is noteworthy, since it provides positive feedback that the CAP-tests are working correctly. Theoretically, unbound $\Delta M_{LRG}$ values are expected to follow the same distribution as their underlying magnitudes; on the other hand, bound $\Delta M_{LRG}$ values deviate from their GR-MFD[32]— because $M_{MAX}$ restricts bound $\Delta M_{LRG}$ values, creating apparently steeper $b$-values (Fig. 1). This expected difference in $\Delta M_{LRG}$ distributions is consistent with the observations from our EW-test separation (Fig. 7). More generally, this biasing phenomenon could potentially explain the anomalously high $b$-values typically associated with earthquakes encountered during HF stimulation[41], since a temporally growing $M_{MAX}$ produces a long and pronounced roll-off at the high-magnitude tail of the GR-MFD.

Furthermore, this finding is significant, since these bound cases are appreciably less hazardous. For example, we can consider a hypothetical situation where the operator approaches a red-light. If $M_{LRG}$ was currently 0.30 M away from the red-light, the next largest event would trigger the red-light in ~30% of unbound cases but only ~4% of the bound cases (Fig. 7). This is especially important considering that earthquake consequences scale exponentially with magnitude[42]. Overall, this hypothetical scenario demonstrates how CAP-tests can be used to quantify magnitude jumps, providing insights into red-light designs and operational decision-making. Correspondingly, we note that operations that encountered significant consequences or were prematurely terminated (e.g., Basel, Pohang, PNR-2) tended to encounter unbound earthquakes[32].

This approach could find near-future use in *post hoc* analyses between stage stimulations – especially in tandem with completion engineering information. Potentially even providing complementary information for identifying hydraulic connectivity between stages. While we have used

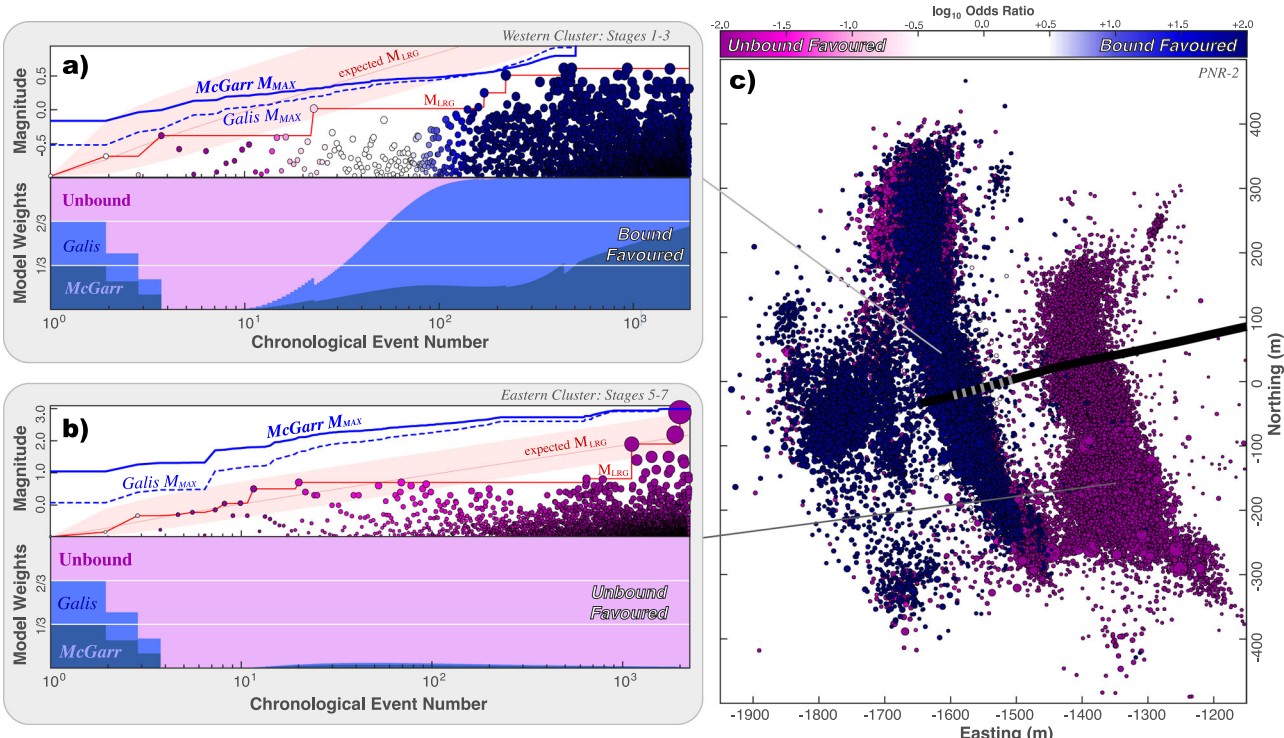

**Fig. 6 | EW-tests applied to PNR-2.** EW-tests are shown for **a** the western-most and **b** eastern-most clusters. The input data includes earthquake magnitudes (circles), the sequence of observed $M_{LRG}$ (red line), alongside the expected $M_{LRG}$ at the 10/50/90 percentiles (red area); two possible injection-based $M_{MAX}$ models are considered (blue lines). Correspondingly, the ensemble weights from the EW-test are shown; the seismic response is spatially partitioned into eastern/western clusters, where stages sequentially transition from bound into unbound (Figs. S25–S26). **c** Map view showing the locations of earthquakes (circles) alongside the PNR-2 well trajectory (black line) and stage locations (grey lines). Throughout the figure, earthquakes are colour coordinated accordingly to their likelihood of being bound or unbound from EW-tests.

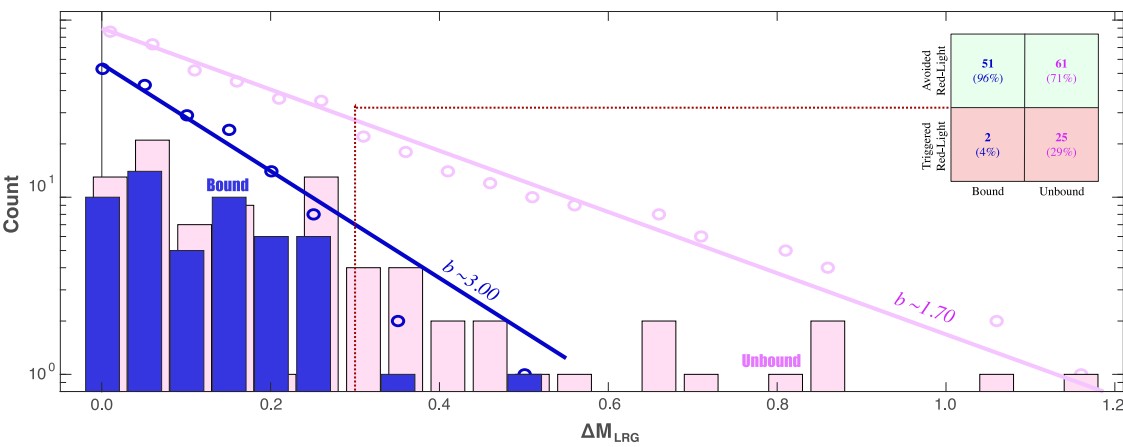

**Fig. 7 | Comparing $\Delta M_{LRG}$ distributions.** Clusters were separated into bound (blue) or unbound (pink) groups using EW-tests. The $\Delta M_{LRG}$ data are plotted as both the cumulative (circles) and non-cumulative (bars) distributions, alongside the GR-MFD fit to the data (solid line). A hypothetical traffic light protocol (dashed line) informs a confusion matrix (inset), that delineates the amount/proportion of bound/unbound events that might trigger the red-light.

CAP-tests to demonstrate this approach on HF and EGS cases, we anticipate this approach will be useful for risk management of other types of induced seismicity (e.g., disposal, mining, reservoir impoundment) as well as tectonic earthquakes. Furthermore, McGarr/Galis predictions of small magnitude $M_{MAX}$ at low volumes suggest the potential to test these models at smaller scales, or use deep catalogues to improve warning times. With further development, this type of approach could be used to quantify logic-tree weights of $M_{MAX}$ within seismic hazard analysis[31] or to inform seismicity forecasts with real-time data. Although, we caution that the potential for a bound-to-unbound transition (Fig. 8) needs to be considered. We

expect CAP-tests to be an important part of future tools that de-risk induced seismicity.

**Stimulation optimisation.** Distinguishing between bound fracture growth and unbound fault reactivation is also significant from the perspective of reservoir optimisation. Stage fluids diverted into faults hinder stimulation efficiency and thus limit reservoir productivity[43]. The use of CAP-tests could be paired with complimentary diagnostic tests attempting to discern the presence of faults[44]. Future operators could use these diagnostic tests to proactively discern stages that are linked to faults,

**Fig. 8 | Interactions between stimulated fractures and fault reactivation during hydraulic fracturing.** A series of HF stages of increasing complexity (text labels) are considered alongside a hypothetical well (black & grey rectangle). In the simple case, the host rock is initially split in tension via stimulated fractures (grey polygons) and may also subsequently reactivate in shear. In the complex case, small pre-existing faults (black polygons) can also be reactivated in shear slip. The clustered case hydraulically connects a series of stages/perforations into a single fracture/fault network. In the last case, stimulated fractures intersect a large fault system that is critically-stressed for slip. By watching the growth of earthquake magnitudes, CAP-tests can discern between bound cases (blue circles) and unbound cases (pink circles).

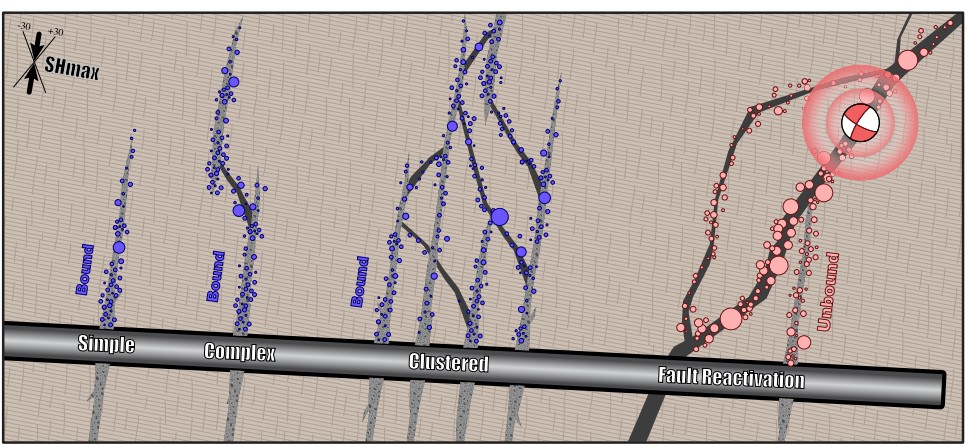

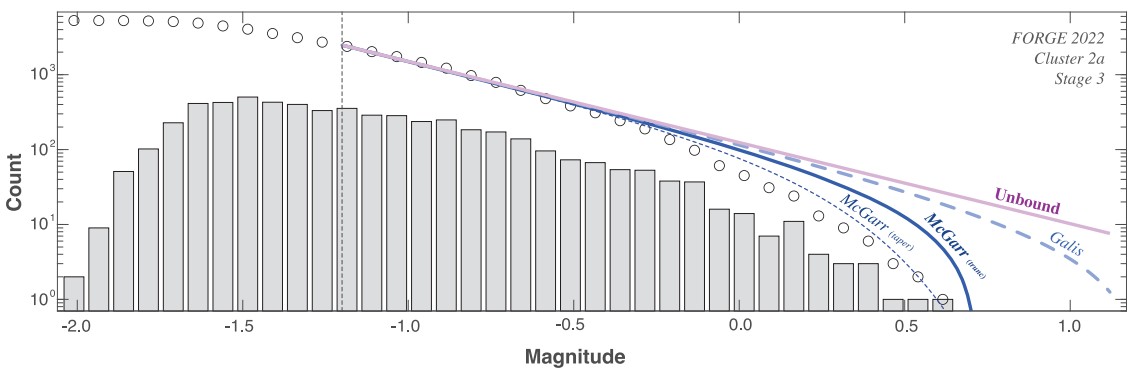

**Fig. 9 | Comparison of expected fits to the GR-MFD.** Data are plotted as both the cumulative (circles) and non-cumulative (bars) distribution, alongside the magnitude-of-completeness Mc (dashed lines). Expected fits to the data based on the unbound (pink line), Galis-like (pale blue dashed line), McGarr-like with a truncated GR-MFD (dark blue solid line), and McGarr-like with a tapered GR-MFD (dark blue dashed line). Data is from stage 3 stimulation at FORGE in 2022 (cluster 2a). The magnitude-of-completeness Mc is also shown (dashed lines).

thus avoiding stimulation into unproductive and seismically risky stages. On the flip side, operators that recognise a bound case could potentially continue injecting closer to the red-light – allowing for greater fracture stimulation, and greater reservoir productivity, which enables further energy development. For example, the next largest event for ~50% of bound cases would trigger a red-light if $M_{LRG}$ was 0.10 M away from the red-light (Fig. 7). Although, this would require further research and validation testing into the nuances and efficacy of CAP-tests (e.g., Text S6-S7). A hypothetical scenario could be a bound-to-unbound transition; for example, if bound fracture growth suddenly encounters a large fault (Fig. 8). Generally, having a strong understanding of the fault architecture and hydraulic connectivity appears to be important for discerning bound cases.

**Physical mechanisms.** The operation at PNR-1z, St1, and initial stages of FORGE/PNR-2 appear to have responded in similar manner to stage stimulation; all have encountered some physical process bounding magnitude growth. There is strong evidence for this from the GR-MFD and CAP-tests, that cannot be reasonably explained by statistical variance (Sections 2–3; Figs. 3–7). This means that there is a measurable impact of finite fault size restricting magnitudes in these cases, albeit with a time dependence from the injection.

Certainly, the spatial extent of stimulation earthquakes increases with time during HF operations, including FORGE[45,46]. This finite extent of the stimulated fractures provides a potential candidate mechanism for restricted magnitude growth. As well, there is growing evidence suggesting that

injection-based processes have restricted at least some stage stimulations at HF and EGS operations[16,47–49] and during laboratory experiments[50]. Correspondingly, our results indicate that either Galis-like or McGarr-like models[26–28] can explain the data, depending on the case. This result also corresponds with the scaling of total seismic moment and total injected volume observed during HF operations in Canada[47]. We note that while Galis-like and McGarr-like are the 'winning' models, there could still be other unknown/untested models that explain the data even better. Future high-resolution studies (aided by CAP-tests) will be able to further elucidate the physics of the bounded fracturing process.

On the other hand, cluster 3 at FORGE and the eastern-most cluster at PNR-2 have responded in a fundamentally different way; there appears to be no (currently discernible) bounds restricting the growth of those earthquake magnitudes. Correspondingly, other studies of PNR-2 have also recognised sequential changes in stage stimulations towards the eastern-most cluster[21,51]. We argue that this likely represents fault reactivation: the fluid/pressure diffusion process triggers slip initiation on a large fault, releasing the tectonic stresses stored over geological timeframes (Fig. 8). The earthquake magnitudes were able to grow unrestricted by any physical limitations. This interpretation is consistent with numerous other large magnitude cases that have also exhibited unbound growth, which appears to be the norm[33]. Further work would be needed to better understand the processes responsible (e.g., fracture complexity, hydraulic connectivity, stress orientations) for creating this bound-to-unbound transition[52].

In this sense, we argue that CAP-tests represent an approach able to empirically discern between triggered or driven induced earthquakes[22,53,54].

Earthquakes induced during bound fracture growth are measurably discernible from those triggered during fault reactivation. This approach will be useful in testing against other cases that appear to be deficient in high magnitude events[55,56].

**Future directions for CAP-tests**. We note that our CAP-tests are limited, in the sense that they can only infer the presence of $M_{MAX}$ and then select the best model among the ensemble of $M_{MAX}$ proposals. If the true model is missing from this ensemble, then it is impossible to select. Here, we discuss the potential for this oversight bias in our study, as well as other errors.

To examine this, we focus on the stage 3 stimulation at FORGE (Fig. 4a; cluster 2a). We use the previously fitted McGarr-like and Galis-like $M_{MAX}$ models to build the expected GR-MFD at the end of catalogue recording. To do so, we use a mixture model that performs a summation of truncated GR-MFDs, with the $M_{MAX}$ temporally varying over the course of injection. This allows us to visualise the expected roll-off that the McGarr-like and Galis-like $M_{MAX}$ ought to have imposed on the data (Fig. 9). The gap between data-and-expectation at high magnitudes provides a qualitative sense of model misfit – further indicating as to why the EW-tests preferred McGarr-like to Galis-like to unbound, respectively.

Despite these efforts, there still remains a misfit gap. On one hand, this gap could be the result of an oversight-bias, which would suggest that a more conservative $M_{MAX}$-volume relationship is still needed. On the other hand, additional physical complexities like limited stimulated rock volume, a distribution of fault/fracture areas, or seismic moment conservation could have contributed to this gap. Our examination here considered truncated GR-MFDs, if we instead consider a mixture of tapered distributions[57], then this gap appreciably closes. We note that tapered GR-MFDs have been introduced as a potential explanation for the aforementioned physical considerations, and are generally viewed as advantageous over truncated distributions[57]. Regardless, CAP-test can accurately infer the presence of this roll-off. Further interpretation between these physical mechanisms will require additional studies and information to make more sophisticated $M_{MAX}$ proposals.

Last, there is the potential for errors lurking within the catalogue. These can be related to differing measurement types, differing catalogue building, or magnitude uncertainties, all of which can appreciably impact interpretation[58]. In light of these errors, we have tried to be careful in our analysis, to consider magnitude dithering, bootstrapping and parameter perturbations (Text S6–S7). We note that the relative importance of $\Delta M_{LRG}$ for discerning $M_{MAX}$ provides future directions for a quality-control prioritisation scheme. In cases where constraining $M_{MAX}$ via CAP-tests are important, quality-control efforts should primarily focus on accurately capturing the $M_{LRG}$ envelope (Fig. 1 & S1).

## Conclusions

We find that our CAP tests can confidently discern cases of bound earthquake magnitude growth during HF stage stimulations at PNR-2 and FORGE (including PNR-1z, ST1, and Soultz-sous-Forêts). This includes an identification of a transition into (problematic) unbound growth, which prematurely terminated operations at PNR-2. We discuss the potential physical mechanisms controlling this binding process. We also discuss the implications for operational risks regarding reservoir optimisation and induced seismicity, including our thoughts on possible mitigation measures guided by CAP-tests. These tests could be used to de-risk future operations, aiding energy development and the transition to a green energy economy.

## Reporting summary

Further information on research design is available in the Nature Portfolio Reporting Summary linked to this article.

## Data availability

The data (and codes) used to derive our results are available online at GitHub (https://github.com/RyanJamesSchultz/CAPfr). The catalogue and hydraulic source datasets for FORGE are also available online (https://gdr.openei.org/). As well, the source injection data for both PNR-2 and PNR1-z are available (https://www.nstauthority.co.uk/regulatory-information/exploration-and-production/onshore/onshore-reports-and-data/preston-new-road-well-pnr2-data-studies/; https://www.nstauthority.co.uk/regulatory-information/exploration-and-production/onshore/onshore-reports-and-data/preston-new-road-well-pnr-1z-data-studies/) PNR source catalogue information is also available online (https://webapps.bgs.ac.uk/services/ngdc/accessions/index.html#item173104). Injection and catalogue source data for Soultz-sous-Forêts (https://episodesplatform.eu/). Catalogue data for Helsinki St1 (https://doi.org/10.5880/GFZ.4.2.2019.001).

## Code availability

The codes (and data) used to derive our results are available online at GitHub (https://github.com/RyanJamesSchultz/CAPfr).

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

## Acknowledgements
We would like to thank Kevin England and Jim Rutledge for engaging conversations about the FORGE stimulation programme that directed the insights and interpretations of this paper. We would also like to thank James Verdon & Tom Kettlety for discussions around the interpretation of PNR-1z and PNR-2 operations. We thank Max Werner for discussions on statistical aspects of the paper. Last, we thank Barnaby Fryer and the anonymous reviewers, whose comments helped improve this manuscript. This work was supported by the Seismogenic Fault Injection Test (SFIT) and De-Risking Enhanced Geothermal Energy Projects (DEEP) grants. SFIT is funded by the Swiss National Science Foundation, under project number TMPFP2_224393. DEEP is subsidised through the Cofund GEOTHERMICA by the Swiss Federal Office of Energy (SFOE), which is supported by the European Union's HORIZON 2020 programme for research, technological development, and demonstration under Grant Agreement #731117.

## Author contributions
R.S. created the CAP-tests, gathered the data, analysed the cases, and wrote the manuscript. B.D., D.K., R.F. and P.M. were involved in the processing and curation of the FORGE catalogue. F.L., P.S., V.R. and L.V. were involved in the collection of FORGE data. F.L. and S.W. provided project management and administration. All authors were involved in manuscript editing and review.

## Funding

## Competing interests
The authors declare no competing interests.
