## [Transparent Peer Review file · Communications Earth & Environment]

The bound growth of induced earthquakes could de-risk hydraulic fracturing

Corresponding Author: Dr Ryan Schultz

Version 0:

Decision Letter:

Dear Dr Schultz,

Please allow us to sincerely apologise once more for the long delay in sending a decision on your manuscript titled "The bound growth of induced earthquakes could de-risk hydraulic fracturing". It has now been seen by 3 reviewers, and we include their comments at the end of this message. They find your work of interest, but some important points are raised. We are interested in the possibility of publishing your study in Communications Earth & Environment, but would like to consider your responses to these concerns and assess a revised manuscript before we make a final decision on publication.

We therefore invite you to revise and resubmit your manuscript, along with a point-by-point that takes into account the points raised. Please highlight all changes in the manuscript text file.

In particular, please ensure that your revised manuscript meets the following editorial thresholds:

* Present robust estimations of Gutenberg-Richter parameters by either recomputing them as requested by Reviewer #3 or compellingly demonstrating that the ones you use are reliable.

* Make explicit the fundamental logical leaps linking earthquake distributions and M_{MAX} .

* Provide a thorough explanation of how this paper represents a significant advance over previously published work including your own.

Please submit your point-by-point responses as a separate file, distinct from your cover letter where you can add responses to the Editors' comments that you do not want to be made available to the reviewers. Word files are preferred. We recommend that any figures, tables or graphs that are included in the to reviewers are also included in the main article or Supplementary Information.

Please use the following link to submit your revised manuscript, point-by-point to the referees' comments (which should be in a separate document to any cover letter), a tracked-changes version of the manuscript (as a PDF file) and the completed checklist:

Link Redacted

We hope to receive your revised paper within six weeks; please let us know if you aren't able to submit it within this time so that we can discuss how best to proceed. If we don't hear from you, and the revision process takes significantly longer, we may close your file. In this event, we will still be happy to reconsider your paper at a later date, as long as nothing similar has been accepted for publication at Communications Earth & Environment or published elsewhere in the meantime.

Please do not hesitate to contact us if you have any questions or would like to discuss these revisions further. We look forward to seeing the revised manuscript and thank you for the opportunity to review your work.

Best regards,

Dr Gareth Roberts
Editorial Board Member
Communications Earth & Environment
orcid.org/0000-0002-6487-8117

Joe Aslin
Deputy Editor
Communications Earth & Environment

EDITORIAL POLICIES AND FORMATTING

- Behavioural and social science
- Ecological, evolutionary & environmental sciences
- Life sciences

Furthermore, please align your manuscript with our format requirements, which are summarized on the following checklist: <https://www.nature.com/documents/commsj-phys-style-formatting-checklist-article.pdf> Communications Earth & Environment formatting checklist

and also in our style and formatting guide <https://www.nature.com/documents/commsj-phys-style-formatting-guide-accept.pdf> Communications Earth & Environment formatting guide .

***** DATA:** Communications Earth & Environment endorses the principles of the Enabling FAIR data project (<http://www.copdess.org/enabling-fair-data-project/>). We ask authors to make the data that support their conclusions available in permanent, publically accessible data repositories. (Please contact the editor if you are unable to make your data available).

All Communications Earth & Environment manuscripts must include a section titled "Data Availability" at the end of the Methods section or main text (if no Methods). More information on this policy, is available at <http://www.nature.com/authors/policies/data/data-availability-statements-data-citations.pdf>.

If a community resource is unavailable, data can be submitted to generalist repositories such as <https://figshare.com/> or <http://datadryad.org/> Dryad Digital Repository. Please provide a unique identifier for the data (for example a DOI or a permanent URL) in the data availability statement, if possible. If the repository does not provide identifiers, we encourage authors to supply the search terms that will return the data. For data that have been obtained from publically available sources, please provide a URL and the specific data product name in the data availability statement. Data with a DOI should be further cited in the methods reference section.

REVIEWER COMMENTS:

Reviewer #1 (Remarks to the Author):

This manuscript concerns a maximum magnitude hazard assessment method in the context of hydraulic fracturing induced seismicity. The authors present a novel statistical approach for assessing whether seismicity is “bound” or “unbound” on the basis of increment of maximum magnitude. Essentially this translates to an assessment of whether or not the magnitude distribution is truncated for large magnitudes. I am not an expert in statistical methods (and so I recommend a further reviewer for these elements); however, in a more general sense this approach – while unlikely to be infallible - seems reasonable to me. An approach like this will have limitations (which is normal – no one is able to predict maximum magnitude perfectly), so it might be beneficial to address certain aspects more explicitly (for example, there is a degree of time required to assess whether or not seismicity behaves as bound or unbound: the efficacy of this method will depend on the magnitude reached before this assessment can be made) and to be clear on potential limitations (what about testing against established failure cases like Basel or Pohang?). Beyond this, and some minor precisions regarding EGS stimulation, I have principally minor comments.

--- Main manuscript ---

35 – Is it really the case that HF operations are targeting pre-existing fractures primarily?

39 – I would argue there is a distinction between hydraulically fracturing an EGS and stimulating an EGS with shear stimulation. I would argue that shear stimulation is reliant on shear induced dilation, with shear displacement meaning that these joints are not really to be considered as fractures (which do not have lateral displacement).

45 – 8 is not the appropriate reference (it is from 2023). Additionally, this is only one mechanism by which injection can cause reactivation.

57 – “still fail to robustly discern” might be construed as a bit of an understatement. A traffic light system was deployed in Pohang (Hofmann et al., 2018 – Geothermal Energy) and Basel (Håring et al., 2008 – Geothermics).

129 – It might be helpful to the reader if shear modulus and the gamma parameter from Galis were defined/explained. By the way, the gamma from Galis depends on permeability which will not be homogeneous in the case of a hydraulic fracturing operation. Additionally, it is a model which considers injection directly onto a fault plane which does not really correspond to the mode-I deformation considered in this work.

157 – Is 34 the correct reference here? It refers to a paper about PNR-2 in the UK. Maybe it is 35 which discusses Soultz? Regarding the finding that Soultz was found to be a bound case, how do you marry that with the occurrence of four earthquakes $ML > 3$ (e.g., Lengliné et al., 2023; GJI)? Was the bound higher than 3? It seems from Figure S34 that Soultz is bound with $M_{Max} \sim 3$.

Is it possible to vectorize the figures?

It seems from Figure 1 that the most readily discernable difference (by eye) between the bound and unbound models is the magnitude distributions (both M and $dMLRG$). The remaining main text figures do not show this for the real test cases (Figure 1 is synthetic data, Figure 7 is all clusters). However, in the supplementary material Figure S17 is quite a nice example for FORGE. One can see that, like for the synthetic case, the magnitude distributions behave differently for the bound and unbound cases (Figure S18-S21). Would it be possible to include something like this in the main text for an individual real case?

Looking at the example cases, it seems that the model first predicts unbound behavior. This makes sense as, even if the seismicity is ultimately bound by a finite length, that length has not been reached yet. Only after a certain period of time/number of events do the models predict bound behavior. It seems therefore that some level of commitment is required before the model can really predict “relative safety” (as you state in the supp mat line 191). For example, certain cases, while ultimately bound and “safe”, do not register as bound until the M_{Max} unbound prediction is already above 2, with an error bar up to ~ 3 (e.g., Figure S34, S36, and to a lesser extent S40). Is it fair to say that, because of this, even the majority of the bound model predictions (in terms of across the time of one operation) will practically be based on a prediction with an “unbound” b -value/seismogenic index? Could you comment on this in terms of practical application?

Could you comment on cases where the frequency magnitude distribution curtails as in your bounded cases but still produces a relatively large event compared to the rest of the catalogue? Would this result in a false bound prediction? E.g., Goertz-Allmann et al., 2017 JGR; Goertz-Allmann and Wiemer, 2013 Geophysics

You note that certain bound cases may begin without the M_{max} being distinguishable (e.g., Figure 2) such that the prediction switches from unbound to bound in time; however, is it also possible that a case has an M_{max} value for a period, but as the footprint of the operation grows, the case changes back to unbound if, at that point, a fault is encountered? How quickly could your model respond to such a case if it already had a large catalogue of “bound-like” $MLRG$ values? You have a section about transitions to unbound growth (section 3), but these do not seem to be transitions strictly speaking but rather different sections of the same well.

Would your model predict unbound behavior for Pohang? It seems this data is available online (doi: 10.5281/zenodo.3529856). Or, otherwise, Basel (which the authors have access to I imagine) since these two cases are probably the most notable in EGS.

--- Supplementary Material ---

Text S8 - Probably Shapiro should be cited if seismogenic index is being used.

I am not sure of the journal's policy, but a number of the references in the supplementary material are not present in the main text.

--- English ---

The level of written English is high. I found no mistakes.

Sincerely,
Barnaby Fryer

Reviewer #2 (Remarks to the Author):

General comment :

The draft article proposed by R. Schultz and co-authors is concerned with the possibility of predicting the maximum magnitude of an earthquake induced by anthropogenic activity. The question is important and still open. The exploitation of "conventional" resources such as oil & gas, or "renewable" resources such as heat from high-energy geothermal energy, only becomes economic once a certain quantity per unit of time has been reached. To obtain this flow from boreholes, it is often necessary to improve some of the properties of the targeted geological medium, most often its permeability. This can be achieved by forced injection of a fluid. The technique of hydraulic fracturing (HF), which consists of creating new fractures by breaking rocks under tension, is commonly used by the oil industry to exploit shales - i.e. in a sedimentary context - using sub-horizontal drilling pads. Using geomechanic concepts, it is possible to calculate a priori the volume of fluid that needs to be injected to produce a new fracture surface with a pre-selected geometry (direction, length). The operation is generally accompanied by a swarm of micro-earthquakes, evidence of leakoff and shearing on pre-existing natural fractures, and sometimes (see figure 8) by a higher-energy event that may exceed an acceptable threshold. This is the situation that must be avoided. The example is taken from the British boreholes PNR1 and 2.

For geothermal energy, which is a more recent activity and is developing more in unconfined basement contexts, the HF technique is being tested on boreholes at the Utah FORGES site and at Helsinki St1 site. But another database (here SSFS in supplementary material) relates to the so-called 'EGS' technique, which recommends 'stimulation' along pre-existing natural fractures, starting from deviated boreholes open over long sections, in which the "HF" hydraulic fracturing pressure levels are not reached, and the volumes injected are not defined a priori by an 'HF' type calculation. Seismicity results from a sliding process on pre-existing fractures, sometimes called Hydro-Shearing, which has a positive irreversible effect on permeability. However, several projects (in Europe and Korea) have been halted because the induced earthquake was too great. The question is: could a study of the characteristics of the swarm of seismic events have made it possible to anticipate and stop an injection process in time?

More generally, the proposed article aims to promote an important piece in a work flow for a TLS "traffic light system" that would be more reliable than the algorithms already proposed. The article is perplexing, as it could be more systematic. As it stands, the sites selected to demonstrate the capabilities seem too well chosen. Several of the co-authors have, for example, worked on Iceland, a country that is ahead of its time in the use of geothermal energy, not "HF", and whose databases could have been (re)used (Cf Broccardo, M.,, Karvournis D., .., Wiemer, S., 2020, <https://doi.org/10.5194/nhess-20-1573-2020>).

The conclusion could be more nuanced. It is insisting on the fact that a necessary condition for de-risking an "HF" operation could be to identify as soon as possible "a unbounded growth process", a point the « CAP » package can do, but will this be sufficient ? An example is the case of SSFS illustrated in the supplementary materials, where the likelihood of « bound growth » was established but accompanied by ruptures with magnitudes greater than 2.7. In the end, wouldn't it be better to cap the fluid volumes involved at a few thousand m³ in all cases, in the sense of Galis, for example? Or shouldn't the authors announce a combination with more quantitative approaches, once a degree of certainty about the likelihood of the situation "Mmax_bounded" is obtained?

The text also leaves the impression that the 'HF' technique would be the main alternative for renewable geothermal energy, which is not yet demonstrated. The authors do not mention that HF technology has already been banned in some countries. Only one plant has been launched since the end of 2024, based on the experiments carried out at Utah FORGE. (FERVO Energy's Cape geothermal project, Utah) and to date there has been no feedback on the long-term validity of the system.

Minor comments throughout the text :

Abstract:

In the abstract line 18 , 20 and 23 , introduction , line 5, 23, etc, and caption figure 8 : the wording 'stimulation' is inappropriate. Hydraulic fracturing « HF » requires an injection pressure above the value of the minimum principal component of the stress tensor, to create a new fracture in tension // at SH_max ... (grey polygon in fig 8). The word "stimulation" has more to do with modifying the properties of a pre-existing fracture network, or even improving its connectivity, which is achieved at lower pressures (the "fault reactivation" case in figure 8). In figure S7, the caption is again imprecise: it would be preferable to call the Helsinki St1, Utah FORGE and SSFS "geothermal project" instead of EGS, and the UK PNR "Oil&gas", and to reserve HF for the case where a reservoir is developed in stages using hydraulic fracturing with proppants (Utah FORGE, PNR1&2) or not (Helsinki ST1).

1 Introduction

This section contains the technical description of the new work flow, which is based on a series of statistical tests for analysing sequences of induced events recorded over time (CAP tests). According to the Gutenberg-Richter 'GR-MFD' law, a maximum potential magnitude is statistically predictable, but remains a function of the number of events in the catalogue. The authors suggest a methodology for testing the likelihood of the existence of a stopping condition (a physical process) that would guarantee in advance a maximum value M_{max} for the magnitude, lower than the 'GR' forecast. They summarise several test cases detailed in parts 2 and 3 and show that this is possible. The best classical model prediction is even obtained, based on cumulative injection volumes (MacGarr et al., 2014 or Galis et al., 2017). On the other hand, the analysis makes it possible to detect in advance so-called "unbound" situations in which the rupture mechanism would become uncontrollable, resembling the reactivation of a fault, with a probable magnitude that would be unacceptable. Line 80: ΔM_{LRG} is somehow converging to 0 under the 'bound' option. This part is clear, but please check caption of figure 1.

Figure 1: The 'upper panel' caption (left/right) does not seem to correspond to the labels (respectively bound/unbound). Also, the ΔM_{LRG} catalogue contains far fewer elements than the M catalogue. The axes on 'middle panel' should transcribe this. 'Bottom panel' confirms how a new value of ΔM_{LRG} can change the balance on the likelihood of the bound/unbound hypothesis. It might be useful to show the point in the series at which the 'certainty' of the 'bound' hypothesis is favoured.

Line 85: Weight EW-tests. There is no clear justification for applying the test to the cumulative catalogue of events from several successive sequences, or to clusters, as in the case of FORGE (Fig. S15 and S17), when the injection site is moved along the borehole. Is the statistical method still valid with small data sets ? In figure S17, please, check caption for cluster 3.

2. Bounded growth

Section 2 analyses situations where the proposed work flow leads to the hypothesis of a mechanism limiting the maximum magnitude ('bound growth stage')

Synthetic scenarios (supplements S5 and S6) show how the Max magnitude can be estimated with confidence, according to the McGarr or Galis models. Two site specific data sets are used in the main body of the text, from shale gas and deep geothermal exploration respectively.

Note : Figure S8 and S11 are not colored according to stage, as stated.

The Helsinki St1 geothermal project (Otaniemi project) from which the processed data originate (see figures 4, S11-S13) was ultimately abandoned for lack of productivity. The multi-stage HF strategy followed (Kwiatek et al. 2022, Kukkonen et al. 2023) to steer the development phase by continuous quantitative hydro-mechanical analysis was successful, since the "administrative" magnitude threshold set at 2 was not exceeded, with a maximum magnitude recorded of 1.9. But the industrial project remains a failure. The new methodology proposed here would prove that the "bound" hypothesis appears to be the most favoured, which an operator could have used to say that the risk was under control. However, the probable next jump ΔM_{LRG} cannot in this case be less than the distance to the red_light, of 0.1.

Do you think that the "proceed as planned" argument would have been convincing enough to continue the test, given that the probability was only significantly in favour of the Galis model after several hundred events and a penultimate jump of magnitude ~1.5?

Question : Figure 4a) does not show the event with Mag 1.9. (Cf as stated in line 97) . Figure supplementary S12 and S13 show 2 events, eventually not falling into the 1.9 bin

Section 2 concludes with a note on the Soultz site, in supplementary material (Figure S27 to S36). These data do not reflect an 'HF' development protocol. Nevertheless, the algorithm identifies the 'bound' hypothesis for the maximum magnitude at all stages of the project's development:

Question: Regarding the choice of the 1993 extract (S27): What exactly is the test illustrated? There were several hydraulic sequences in 1993, with up to 10,000 events recorded. Here, there are only a hundred or so, even though there was a massive injection test that ended with a real opening mechanism, oriented in the direction of the major stress (Cornet et al., 2007)?

Question: The SSFS test, dated 2000, is the stimulation of the first deep well GPK2, followed by wells GPK3 in 2003 and GPK4 in 2004 and 2005. During each operation, larger volumes were mobilised (25,000, 37,000, 9,000 and 12,000 m³) and the maximum magnitude recorded exceeded the value of 2.5 at least once. Could you comment on the b_{values} shown in the figures for the SSFS test, which are all less than 1? Why are the values lower than those published (Charley et al., 2007) for operations on the deep boreholes (S29, S31, S33, S35) at the Soultz sous Forêts site?

In table 1 S1, 4 times a MacGarr model is considered (and the likelihood of a bounded behavior favored), whereas figure S32 for SSFS 2003 seems still at risk, with magnitudes of 2.7 recorded at the end of the operation. Lastly, b-values <1 are usually associated with fault structures, which tend to favour an unbound mode. Further on in section 3, the analysis of cluster 3, stage 7-10, on Utah FORGE in 2024 will result in a "fault" type rupture, with a b_value of 1.8. Is this consistent?

The article would benefit from a more detailed account of this SSFS case in the main body of the text, as it is the only one that does not involve the "HF" technique *stricto sensu*. The SSFS project has been in operation for more than 10 years, with no significant contribution from the last very low-productivity well.

3 The transition to unbound growth

This is a general comment: in all the cases presented, the option of an unbounded model is favoured up to the arrival of around ten events. Do you have any experience of a case with a M_{max} that was evaluated as probably "bound" up to a certain number of events (and therefore a certain volume injected) and then switched back to an "unbound" mode? This is certainly conceivable, since a number of major events have occurred after the injection was stopped, or even several months later (Cf site de Bales, Häring et al., 2008, Catalli ... Wiemer, 2013 doi:10.1029/2012GL054147) Could there be a bias linked to the fact that an "HF" test is necessarily limited in volume?

Question: If it were the case that the regulation only authorised a maximum volume for a stimulation, in the "EGS" sense, then would this purely statistical tool prefiguring a TLS be useful, since it seems to start every time by switching to the "unbound" mode?

In the case of 'HF' operations for shale treatments, illustrated by figures 6 and S22, PNR2-stage 5-7 test, carried out in 2024, it is difficult to understand what shutdown measures could have been taken before the major event that occurred 3 days after the shut-in (see line 204) for only 2,500 m³ used, since the "HF" treatment was identified by the work flow as most likely to have been 'unbound' very early on: Would it be appropriate to illustrate this case with an abscissa relative to volume or time and in // the evolution of b-value?

4 Implications

Section 4 could address another major point. The question of whether the 'stimulation' operations analysed were hydraulically conclusive is not envisaged.

Would a continuously controlled operation with a growth in microseismic activity classified as "bound" necessarily be conclusive from a hydraulic point of view? The case of Helsinki St1 proves that the 'HF' treatment did not achieve the objectives in the economic sense.

Similarly, in the Utah FORGE project, the success of the 2022 operation (stage 3, figure 5a) remains. On the basis of the 2024 operations, stage 7-10, the last cluster 3 is identified by the new methodology as 'unbound', i.e. classified as at risk, and the operation could/should have been stopped earlier. The maximum magnitude recorded was 2.1. However, the subsequent hydraulic circulation and tracer tests (reported elsewhere by one of the co-authors) showed that the flows obtained over the entire length of the borehole (all stages) were unevenly distributed, and that the zone relating to cluster 3 played a major role in the final hydraulic efficiency obtained. Non-constant spacing and irregular contribution, are known conditions likely to promote early thermal breakthrough.

Note Figure 8: The illustration of the "complex" situation is strange, because maintaining conditions in tension after a shear relay is not very realistic, unless the first grey polygon continues to extend at the same time. The 'clustered' case is also awkward. There is very little chance that the strongest magnitude will be on a grey polygon propagating in tension.

Reviewer #3 (Remarks to the Author):

This paper addresses the problem of determining the maximum magnitude (M_{max}) of induced seismicity. In my review, I will focus on the statistical part of this paper.

I want to start with the fact that several seismological papers assert the extreme difficulty in the estimation of M_{max} , e.g.: Holschneider, M., Zöller, G., & Hainzl, S. (2011). Estimation of the maximum possible magnitude in the framework of a doubly truncated Gutenberg–Richter model. *Bulletin of the Seismological Society of America*, 101(4), 1649-1659. Zöller, G., & Holschneider, M. (2016). The earthquake history in a fault zone tells us almost nothing about M_{max} . *Seismological Research Letters*, 87(1), 132-137.

Zöller, G., Holschneider, M., & Hainzl, S. (2013). The maximum earthquake magnitude in a time horizon: Theory and case studies. *Bulletin of the Seismological Society of America*, 103(2A), 860-875.

In this work, it seems that with some induced seismicity catalogs, it is possible to infer the existence of a maximum magnitude. However, to properly infer this existence, the other fundamental parameters of the Gutenberg-Richter law must be robustly estimated: the magnitude of completeness (M_c), and the b-value.

I found some important flaws regarding these Gutenberg-Richter parameters:

1) In the Supplementary Material, the authors state: "The M_c is simply evaluated by examining the peak bin of the non-cumulative GR-MFD and then setting conservatively." Such a type of estimation is not proper in my opinion, given the importance of M_c in the b-value estimation and the count of the number of available events. The authors should use modern techniques, such as those based on the exponentiality of data:

Herrmann, M., & Marzocchi, W. (2021). Inconsistencies and lurking pitfalls in the magnitude–frequency distribution of high-resolution earthquake catalogs. *Seismological Society of America*, 92(2A), 909-922.

Taroni, M. (2023). Estimating the magnitude of completeness of earthquake catalogs using a simple random variable transformation. *The Seismic Record*, 3(3), 194-199.

Moreover, the authors did not investigate the short-term incompleteness problem, which also affects induced seismicity catalogs, and the corresponding b-value estimation.

2) According to Table S1, the estimated b-values span from 0.53 to 1.88. This is a very large variability, which could be caused by an improper estimation of the M_w magnitude for such small events, or an incorrect estimation of the magnitude of completeness.

3) In this work, another way to model the right tail of the Gutenberg-Richter distribution, i.e., the Tapered Gutenberg-Richter model, is completely neglected. This model is widely used in statistical seismology studies and gives some advantages with respect to M_{max} in its estimation. See, e.g.:

Kagan, Y. Y. (2002). Seismic moment distribution revisited: I. Statistical results. *Geophysical Journal International*, 148(3), 520-541.

Kagan, Y. Y., & Schoenberg, F. (2001). Estimation of the upper cutoff parameter for the tapered Pareto distribution. *Journal of Applied Probability*, 38(A), 158-175.

To conclude, I suggest that the authors recompute all the M_c using a more suitable method, check the reliability of the magnitudes used in their calculation, and consider the Tapered Gutenberg-Richter model (at least for some comparison test).

Since these points are fundamental to understanding the reliability of their results, I suggest a major revision for this paper.

** Visit Nature Portfolio's author and referees' website at www.nature.com/authors for information about policies, services and author benefits**

Communications Earth & Environment is committed to improving transparency in authorship. As part of our efforts in this direction, we are now requesting that all authors identified as 'corresponding author' create and link their Open Researcher and Contributor Identifier (ORCID) with their account on the Manuscript Tracking System prior to acceptance. ORCID helps the scientific community achieve unambiguous attribution of all scholarly contributions. You can create and link your ORCID from the home page of the Manuscript Tracking System by clicking on 'Modify my Springer Nature account' and following the instructions in the link below. Please also inform all co-authors that they can add their ORCIDs to their accounts and that they must do so prior to acceptance.

If you experience problems in linking your ORCID, please contact the Platform Support Helpdesk.

Version 1:

Decision Letter:

Dear Dr Schultz,

Your revised manuscript titled "The bound growth of induced earthquakes could de-risk hydraulic fracturing" has now been seen by our reviewers, whose comments appear below. In light of their advice we are delighted to say that we are happy, in principle, to publish a suitably revised version in *Communications Earth & Environment*.

We therefore invite you to revise your paper one last time to address the remaining concerns of our reviewers. At the same time we ask that you edit your manuscript to comply with our format requirements and to maximise the accessibility and therefore the impact of your work.

EDITORIAL REQUESTS:

*****Please take care to match our formatting and policy requirements. We will check revised manuscript and return manuscripts that do not comply. Such requests will lead to delays. *****

Please outline your to each request in the right hand column. Please upload the completed table with your manuscript files as a Related Manuscript file.

SUBMISSION INFORMATION:

OPEN ACCESS:

Communications Earth & Environment is a fully open access journal. Articles are made freely accessible on publication. For further information about article processing charges, open access funding, and advice and support from Nature Portfolio, please visit <https://www.nature.com/commsenv/open-access>

Link Redacted

Best regards,

Dr Gareth Roberts
Editorial Board Member
Communications Earth & Environment
orcid.org/0000-0002-6487-8117

Alireza Bahadori, PhD
Senior Editor
Communications Earth & Environment
Consulting Editor
Communications Sustainability

REVIEWERS' COMMENTS:

Reviewer #1 (Remarks to the Author):

Line numbers correspond to the tracked changes version of the manuscript.

My comments have been mostly answered. I only have minor comments remaining, although I think it is important for the other reviewers to also be satisfied.

Regarding the comment about the discernible differences between bound and unbound cases by eye, the authors would like to emphasize the difference in distributions between the two cases. Currently they state this as (line 295), "Theoretically, unbound dMLRG values are expected to follow the same distribution as their magnitudes; on the other hand, bound dMLRG values deviate from their GR-MFD [32] – because MMAX restricts bound dMLRG values, creating apparently steeper b values (Figure 1)." I think this is quite clear. Maybe the part "as their magnitudes" could be more explicit; something like "as the earthquake magnitudes on which they are based".

Regarding the comment about a rolled-off frequency magnitude distribution that still manages to produce a large event (e.g., Goertz-Allmann and Wiemer, 2013 Geophysics), and looking at some examples from the supplementary material (Figures S9-10, S12-13, S17-21), as you say it does seem like bound cases do have this roll of that unbound cases do not have. For that reason, it seems likely to me that Figure 15 of Goertz-Allmann and Wiemer might cause problems for the model. This seems to be related to certain concerns of reviewer 3.

Line 349 it seems like this new phrase is not a complete sentence.

Concerning Figure 7, I suppose looking at dMLRG means that there are much fewer data points than when one makes a traditional frequency magnitude distribution. How many dMLRG points are needed to provide confidence in the b value they yield? Maybe this also ties in to the previous comment about bound cases becoming unbound if they run into a fault (although I see the new text regarding this case).

Reviewer #3 (Remarks to the Author):

In this revised version of the paper, the quality of the work is increased, and the authors properly replied to most of my comments. Thus, I think that this paper is now ready for publication.

** Visit Nature Portfolio's author and referees' website at <http://www.nature.com/authors> for information about policies, services and author benefits**

Dear Editors of Communications Earth & Environment,

Thank you for taking the time to handle our manuscript during its revision process. We are happy to hear constructive and thorough responses from the reviewers – we agree with their assessments that this work nicely contributes to a difficult and open-ended problem in seismology that has important scientific and practical implications. Generally, the critiques focused on clarifying terminology, better explaining our methods, highlighting results from other cases, and providing more rigorous assessment of the magnitude-of-completeness. We note that many of the comments were open-ended curiosity-based questions, which is exciting to see the reviewer’s meaningfully engaging with our work!

We have endeavored to fully address these concerns through new analyses, new/revise figures, additional paragraphs, and sentence/wording revisions that better outline the details of this work; as well, a few new references have been added, in response to these changes. The most substantial changes are a new discussion section that addresses multiple reviewer comments. Finally, we have also re-read the paper with fresh eyes to catch any lingering grammatical or typographic mistakes. We feel that these changes to the revised version adequately address the reviewer’s comments, making the manuscript suitable for publication – we hope that you agree.

*On the following pages are our itemized responses to the reviewer’s comments. The original comments are in **bold-face**, our responses are in italics, and callouts to the original text are in “blue-face.”*

Thank you again,

- Ryan & co-authors

Editors:

Please allow us to sincerely apologise once more for the long delay in sending a decision on your manuscript titled "The bound growth of induced earthquakes could de-risk hydraulic fracturing". It has now been seen by 3 reviewers, and we include their comments at the end of this message. They find your work of interest, but some important points are raised. We are interested in the possibility of publishing your study in Communications Earth & Environment, but would like to consider your responses to these concerns and assess a revised manuscript before we make a final decision on publication.

We therefore invite you to revise and resubmit your manuscript, along with a point-by-point response that takes into account the points raised. Please highlight all changes in the manuscript text file.

We would like to thank the editors again for handling our manuscript. Furthermore, we are grateful to hear that our article is being considered for publication! In summary, the editors would like to see three quality thresholds are met before reaching a final decision.

For each of these thresholds, we provide a brief explanation as to how we've met these criteria. We note that these descriptions only cover the high-level response. More in-depth details can be found in our thorough responses to the respective reviewer comments. In those responses, we more deeply show how we're meeting these three quality thresholds.

In particular, please ensure that your revised manuscript meets the following editorial thresholds:

*** Present robust estimations of Gutenberg-Richter parameters by either recomputing them as requested by Reviewer #3 or compellingly demonstrating that the ones you use are reliable.**

For this point we have opted to do both – to be exhaustive on this issue.

First, we have used a more robust selection process for M_c . These new M_c values do not impact the results of our study; in part because the new/optimal values are close to our prior values.

Furthermore, we have shown that if we perturb those M_c values by ± 0.2 , it also does not change the results/interpretations of our study. This range of values includes the prior choice of M_c .

*** Make explicit the fundamental logical leaps linking earthquake distributions and M_{MAX} .**

We are more than happy to do this. We have re-read our paper with a refreshed/critical eye for explicitly explaining all of the methods/concepts. This is in addition to the responses to Reviewer #3's comments regarding truncated/tapered GR-MFDs. For this point we have added a new analysis and figure (Section 4.4 & Figure 9). Many other relevant manuscript changes also appear in the supplementary sections.

For this point we would be grateful for a small clarification, if possible. In part because the reviewers didn't point out many logical leaps in this regard (with the exception of truncated/tapered distributions). To the contrary, most seemed to follow the methods well, even describing them back to us accurately. That said, if there's any specific points the editors stumbled on, we'd be happy to revise it. In fact, any direction from fresh eyes would be greatly appreciated, as we are familiar with the approach, and might be myopic to what exactly we're missing.

*** Provide a thorough explanation of how this paper represents a significant advance over previously published work including your own.**

Yes, we are happy to provide this!

As Reviewer #3 points out, inferring an M_{MAX} from a catalogue is an extremely difficult problem. Because of this, seismology has a methodological blind spot. This problem is also unfortunate, as many authors have suggested volume-based M_{MAX} models (Refs #26-28, to name a few). To date, these models have effectively remained untested, because this blind spot prevents empirical verification. Even more unfortunate yet, if these models do exist, then they have the potential to significantly reduce the amount of seismic hazard anticipated during an operation.

In prior works [Schultz, 2024], we reframed this M_{MAX} inference problem to examine ΔM_{LRG} instead of just M . This is expected to significantly increase the resolvability of M_{MAX} from a catalogue and even discriminate between best fitting models. We then tested this approach on a handful of high-profile cases that skirt around McGarr's upper limit – none of these cases had any indication of being affected by a volume-based M_{MAX} , they were all unbound.

What is new in this paper is that we have found bound cases of induced seismicity. Because of the aforementioned blind spot, this has not been accomplished before. We then show that we can discriminate between bound/unbound and even identify a bound-to-unbound transition during an operation. The cases that we've found so far, seem to have similarities to each other – all of them have used stimulation techniques. Because of that, we have made an interpretation that fracture growth is bounding the magnitude growth, although verifying this interpretation will require further study.

These findings are significant (in our opinion) for three reasons. First, this is novel in the sense that we have discovered/identified something that was suggested to exist, but never demonstrated to this level of rigour. It took a special methodology to accomplish that. Second, this is impactful for future research, because it gives a tool that can identify bound/unbound cases: the community can start rigorously testing for models, discarding unverified models, and maybe even finding cases that point to theoretical gaps. Third, this opens avenues for practical risk management: a tool that can discern between unbound/bound cases also discerns more/less hazardous cases.

We have revised the introduction to better explain this methodological blind spot, novelty of our work, and its significance.

Reviewer #1:

This manuscript concerns a maximum magnitude hazard assessment method in the context of hydraulic fracturing induced seismicity. The authors present a novel statistical approach for assessing whether seismicity is “bound” or “unbound” on the basis of increment of maximum magnitude. Essentially this translates to an assessment of whether or not the magnitude distribution is truncated for large magnitudes. I am not an expert in statistical methods (and so I recommend a further reviewer for these elements); however, in a more general sense this approach – while unlikely to be infallible - seems reasonable to me. An approach like this will have limitations (which is normal – no one is able to predict maximum magnitude perfectly), so it might be beneficial to address certain aspects more explicitly (for example, there is a degree of time required to assess whether or not seismicity behaves as bound or unbound: the efficacy of this method will depend on the magnitude reached before this assessment can be made) and to be clear on potential limitations (what about testing against established failure cases like Basel or Pohang?). Beyond this, and some minor precisions regarding EGS stimulation, I have principally minor comments.

We would like to thank the reviewer for their suggestions, as well as their generally favourable outlook on this manuscript. We agree that this paper is a novel assessment of an important problem. In our pages below, we outline detailed responses to each of the reviewer’s comments – including parts about limitations of the methods and extension to other failure cases.

--- Main manuscript ---

35 – Is it really the case that HF operations are targeting pre-existing fractures primarily?

In this sentence we don’t mean to imply that the fractures are necessarily pre-existing – they could be either newly stimulated, or pre-existing and then stimulated. Similar to the reviewer’s following distinction point on hydroshear/hydrofrack, we have modified the sentences here for clarity.

39 – I would argue there is a distinction between hydraulically fracturing an EGS and stimulating an EGS with shear stimulation. I would argue that shear stimulation is reliant on shear induced dilation, with shear displacement meaning that these joints are not really to be considered as fractures (which do not have lateral displacement).

We agree with the reviewer on this point (and a similar point raised by reviewer #2). To rectify this point to future readers, we have added a new sentence on lines 37-39 that clarifies our definition of stimulation encompasses both tensile failures and shear failures. That sentence now reads as “This stimulation process can either propagate failures in tension or in shear; these failures can happen on either pre-existing fractures or newly created fractures”.

45 – 8 is not the appropriate reference (it is from 2023). Additionally, this is only one mechanism by which injection can cause reactivation.

On this point, we would kindly request some clarification from the reviewer, as we are not exactly sure as to what exactly is (potentially) inappropriate here. If we have misunderstood something, please let us know and we will correct it.

The cited paper reviews a series of mechanisms that can induce earthquakes, one of which is an increase of pore pressure. We cited this paper, as we are describing how induced seismicity can occur in this sentence. The date of reference #8 is also from 2023 (<https://doi.org/10.1038/s43017-023-00497-8>), which is the publication date we have written in our manuscript’s references.

57 – “still fail to robustly discern” might be construed as a bit of an understatement. A traffic light system was deployed in Pohang (Hofmann et al., 2018 – Geothermal Energy) and Basel (Häring et al., 2008 – Geothermics).

We 100% agree with the reviewer on this point! There is still a clear need for tools to de-risk these operations, as well as a better fundamental understanding of the relevant geophysical processes.

For this point, we have now added another brief point to this sentence that reiterates the detrimental consequences that have been encountered at these cases.

Note that we haven’t added the citation to the Hofmann paper, because we already cite Grigoli for the Pohang case (Ref #10). For the Häring paper, we have already cited this work in the manuscript previously (see Ref #9).

129 – It might be helpful to the reader if shear modulus and the gamma parameter from Galis were defined/explained. By the way, the gamma from Galis depends on permeability which will not be

homogeneous in the case of a hydraulic fracturing operation. Additionally, it is a model which considers injection directly onto a fault plane which does not really correspond to the mode-I deformation considered in this work.

On this point we have been intentionally ambiguous. The reason why is technical: our model isn't necessarily testing for the Galis model per se. Rather, we are just testing for an M_{MAX} model that has a $V^{3/2}$ volume proportionality. One explanation for that observation could be the Galis model. On the other hand, it could also be something else – perhaps an undiscovered model that is relevant for mode-I failures?

In fact, this is part of what makes CAP-tests exciting (in my opinion)! We can start narrowing down cases that actually have evidence for being bounded. This can give the seismological community feedback: we can start focusing on just those bound cases then and then discard the unbound ones that are inapplicable (for this kind of study). After that, we can start examining to see where the theory might be deficient (maybe we find mode-I $V^{3/2}$ cases, meaning we're missing that M_{MAX} model). This should have a nice focusing effect for future research that was previously unavailable.

*To address the reviewer's point, while also respecting this nuance, we have now mentioned that we are testing for *Galis-like* models, instead of Galis models throughout the text.*

157 – Is 34 the correct reference here? It refers to a paper about PNR-2 in the UK. Maybe it is 36 which discusses Soultz? Regarding the finding that Soultz was found to be a bound case, how do you marry that with the occurrence of four earthquakes $ML > 3$ (e.g., Lengliné et al., 2023; GJI)? Was the bound higher than 3? It seems from Figure S34 that Soultz is bound with $M_{Max} \sim 3$.

We thank the reviewer for their attention to detail! This is a typo, we meant to cite the Dorbath paper (i.e., ref #37) instead. This is corrected now.

To the reviewer's point, having events of those sizes is still predicted by the M_{MAX} models – if enough fluid goes in eventually larger events are possible. So, the results are consistent with the data. It's just that bound cases will have a restriction on the size of events that unbound cases do not.

Is it possible to vectorize the figures?

All of our figures are vectorized PDF files that we have ready for the publication/proofing process. That said, what's shown in this manuscript are just PNG screen captures of those vectorized files – to keep the manuscript file size small. We did also do a double-check of the PNG figures in the paper, they appear visible to our eyes.

If the reviewer had a particular figure they wanted to scrutinize more, we would be happy to provide a higher resolution update. Or if there is text, or some other detail, that is too small to see, we'd be happy to adjust. Just let us know, please and thank you!

It seems from Figure 1 that the most readily discernable difference (by eye) between the bound and unbound models is the magnitude distributions (both M and dMLRG). The remaining main text figures do not show this for the real test cases (Figure 1 is synthetic data, Figure 7 is all clusters). However, in the supplementary material Figure S17 is quite a nice example for FORGE. One can see that, like for the synthetic case, the magnitude distributions behave differently for the bound and unbound cases (Figure S18-S21). Would it be possible to include something like this in the main text for an individual real case?

This is an interesting point raised by the reviewer. We agree that showing this information to the reader could help to explain the concepts better while also giving more justifications that show the methods are working correctly.

To address this, we have added a new discussion (Section 4.4) and figure (Figure 9). There we show the fits of the unbound, Galis-like, and McGarr-like models to the data. This makes it clear that these volume-based M_{MAX} models produce a roll-off to the GR-MFD and visualizes why McGarr-like is the best in this case.

We also use this analysis to respond to Reviewer #3 regarding their tapered GR-MFDs comment.

**On this point we also have a request for the reviewer* (if they would be so kind).*

In our original writing we wanted to demonstrate the point the reviewer was getting across in another section (i.e., Section 4.1). Most specifically focusing around the third paragraph (i.e., lines 294-302) and Figure 7. Part of the point we're (trying to) make there is exactly what the reviewer is asking for: that if

we use CAP-tests to separate the clusters into bound/unbound interpretations, then the distributions of those ΔM_{LRG} differs (Figure 7) in exactly the way the theory predicts (Figure 1). Did this point come across to the reviewer?

We have revised this paragraph to try and be clearer on this point, as we earnestly want other readers to recognize this. If the reviewer has any suggestions for how we could have made this point more eloquently, we are open to suggestions.

Looking at the example cases, it seems that the model first predicts unbound behavior. This makes sense as, even if the seismicity is ultimately bound by a finite length, that length has not been reached yet. Only after a certain period of time/number of events do the models predict bound behavior. It seems therefore that some level of commitment is required before the model can really predict “relative safety” (as you state in the supp mat line 191). For example, certain cases, while ultimately bound and “safe”, do not register as bound until the MMax unbound prediction is already above 2, with an error bar up to ~3 (e.g., Figure S34, S36, and to a lesser extent S40). Is it fair to say that, because of this, even the majority of the bound model predictions (in terms of across the time of one operation) will practically be based on a prediction with an “unbound” b-value/seismogenic index? Could you comment on this in terms of practical application?

The reviewer is correct on the data “appearing unbound first” point. To elaborate, we first describe two things relevant for discerning a bound case (and why it always starts as unbound). Note that these discussions are primarily focused on the relevance for EW-tests.

The first is related to the fact that we’re using AIC/BIC scores. These penalize more complex models: the unbound model is the simplest variant, so it always starts as the winning model when data is scarce. Enough ‘good’ data needs to be collected before we can discern the true model. If there is good data, then we usually only need tens of events before being able to discern between bound/unbound cases. We also mention these points in Section S4.2 on lines S265-S274. As an example, the described effect can be seen in all of the EW-test plots in the period of the first 1-20 events recorded.

The second part is related to what constitute ‘good’ data. This is why we introduced the degree-of-truncation metric δM_{LRG} . Our tests indicate that the M_{LRG} needs to be within ~0.5 units of M_{MAX} before anything can be said about M_{MAX} . Note that this is effect is what the reviewer described in start of this comment. Furthermore, a good example of this effect can be seen in the synthetic example of Figure S4

and described in Section S4.2 on lines S275-S284. As well, we describe the theoretical expectation of this in Section S2.2 on lines S92-S98. The details here have strong overlap with a point from Reviewer #3.

To more directly answer the reviewer's comment now. In the tectonic M_{MAX} case, practically speaking, yes, we are at the mercy of the situation. For example, if a red-light was set at M4 and there was a fault capable of hosting M5 events, then this would appear as unbound right up until (and after) the red-light event. However, this isn't necessarily a failing of the CAP-tests, these results will still accurately inform the operator to treat the sequence as unbound, which is correct for the relevant magnitude range and potential M_{MAX} .

However, this difficulty is not necessarily the case for the McGarr-like and Galis-like M_{MAX} models. These models predict small M_{MAX} magnitudes at small injection volumes, with M_{MAX} growing alongside injection. In this sense, we have an M_{MAX} that can be close to the M_{LRG} envelope. This difference also explains why the synthetic EW-tests more quickly discern the McGarr-like/Galis-like models (Figure S5 & S6) than the tectonic model (Figure S4). Said another way, McGarr-like and Galis-like models have made predictions that should be readily testable on data, even at smaller magnitudes.

The point above then gives direction for catalogue improvements. In testing for McGarr/Galis M_{MAX} , having access to small magnitude information should be helpful for discerning M_{MAX} earlier. We now mention this point in Section 4.4.

Based on this comment, we've made a series of small clarifications to the supplements. As well, we added a sentence to the main text's discussion to explicitly mention our response points (lines 328-330). Thanks to the reviewer for prompting this discussion.

Could you comment on cases where the frequency magnitude distribution curtails as in your bounded cases but still produces a relatively large event compared to the rest of the catalogue? Would this result in a false bound prediction? E.g., Goertz-Allmann et al., 2017 JGR; Goertz-Allmann and Wiemer, 2013 Geophysics

This is another good point raised by the reviewer. It's also a difficult question to give a definitive answer to.

If the data behaves according to all the assumptions, then the synthetic tests give answers here (Section S4; Figures S3-S6). Effectively, the confidence is going to be the metric to focus on, as it is the chance that we've got the bound/unbound interpretation wrong. In most cases we have confidences somewhere around 99% or 100-to-1, so this means there's a ~1% (or less) chance an event could fall outside of our expectation.

That said, real-life data doesn't have to follow our assumptions. There could be errors in the measurements of the catalogued magnitudes, or interactions with complicated structures that defy point-process assumptions. We've tried to be as thorough as possible in accounting for these kinds of errors. For example, we dither our magnitudes, consider a range of magnitudes-of-completeness, errors on b -values (Section S5), and have even substituted cases to use entirely different catalogues (Section S6).

For the specific cases pointed out [Goertz-Allmann et al., 2013; 2017], it would be difficult to say anything for certain, off the top of my head. It's true that bound cases will need to have some sort of roll-off. This is why our first steps are to find cases that look like this (i.e., our simple-test) in pre-screening for boundedness. However, just because the GR-MFD shows some roll-off, doesn't necessarily mean it is bound. Furthermore, there are multiple different ways that the dataset could be bound (e.g., tectonic, McGarr-like, Galis-like, or even some as of yet undiscovered model). The CAP-tests would have to be run on that specific case, to start making stronger statements. For example, by making hypotheses about if those cases are bound, exploring if they're statistically significant, and discerning which M_{max} models explain the data best.

You note that certain bound cases may begin without the M_{max} being distinguishable (e.g., Figure 2) such that the prediction switches from unbound to bound in time; however, is it also possible that a case has an M_{max} value for a period, but as the footprint of the operation grows, the case changes back to unbound if, at that point, a fault is encountered? How quickly could your model respond to such a case if it already had a large catalogue of "bound-like" MLRG values? You have a section about transitions to unbound growth (section 3), but these do not seem to be transitions strictly speaking but rather different sections of the same well.

This is another good point raised by the reviewer. We are aware of this possibility. It's partly why we drew Figure 8 the way it is. So that there's some 'foreshadowing' that the unbound case could have started with something that was originally bound. We've now made this nuance explicitly clear with a newly added sentence on lines 333-335.

Thus far, we haven't encountered a case like this, so we're not able to definitively comment. Finding these cases, and then performing synthetic tests could be the work of future follow-up studies, though!

Would your model predict unbound behavior for Pohang? It seems this data is available online (doi: 10.5281/zenodo.3529856). Or, otherwise, Basel (which the authors have access to I imagine) since these two cases are probably the most notable in EGS.

This is an important point raised by the reviewer. The short answer is yes, and we've examined both of those cases already. They were both found to behave in an unbound manner.

This was a finding in the prior work that first described the methodology [Schultz, 2024]. In fact, all of the high-profile cases of induced seismicity there were found to be unbound (Pohang, Basel, Paralana, Cooper Basin, Youngstown, Peace River, Paradox Valley, Guy Greenbrier, Pawnee, Prague). On the other hand, this manuscript documents, for the first time, a verified discovery of bound cases.

We did try to make this point in our manuscript (lines 320-322), as this helps to build evidence for the practical implication around risk management: that the unbound clusters tend to correspond with problematic cases. That said, on this point we could be clearer. We have revised the wording in this paragraph for clarity. As always, we welcome further suggestions here.

--- Supplementary Material ---

Text S8 - Probably Shapiro should be cited if seismogenic index is being used.

This is a fair point, raised by the reviewer. We've added a citation to Shapiro's seminal paper [2010] on the topic. Our apologies for this oversight!

I am not sure of the journal's policy, but a number of the references in the supplementary material are not present in the main text.

We note that the inclusion of supplementary references is standard for many journals. Specific to Communications Earth & Environment, we note that other published articles have handled supplementary references in a similar fashion to our approach. That said, if required, we can revise.

--- English ---

The level of written English is high. I found no mistakes.

Sincerely,

Barnaby Fryer

We would like to thank Barnaby Fryer again for their time and effort taken in reviewing our paper – it is very much appreciated!

Reviewer #2:

The draft article proposed by R. Schultz and co-authors is concerned with the possibility of predicting the maximum magnitude of an earthquake induced by anthropogenic activity. The question is important and still open. The exploitation of “conventional” resources such as oil & gas, or “renewable” resources such as heat from high-energy geothermal energy, only becomes economic once a certain quantity per unit of time has been reached. To obtain this flow from boreholes, it is often necessary to improve some of the properties of the targeted geological medium, most often its permeability. This can be achieved by forced injection of a fluid. The technique of hydraulic fracturing (HF), which consists of creating new fractures by breaking rocks under tension, is commonly used by the oil industry to exploit shales - i.e. in a sedimentary context - using sub-horizontal drilling pads. Using geomechanic concepts, it is possible to calculate a priori the volume of fluid that needs to be injected to produce a new fracture surface with a pre-selected geometry (direction, length). The operation is generally accompanied by a swarm of micro-earthquakes, evidence of leakoff and shearing on pre-existing natural fractures, and sometimes (see figure 8) by a higher-energy event that may exceed an acceptable threshold. This is the situation that must be avoided. The example is taken from the British boreholes PNR1 and 2.

For geothermal energy, which is a more recent activity and is developing more in unconfined basement contexts, the HF technique is being tested on boreholes at the Utah FORGES site and at Helsinki St1 site. But another database (here SSFS in supplementary material) relates to the so-called ‘EGS’ technique, which recommends ‘stimulation’ along pre-existing natural fractures, starting from deviated boreholes open over long sections, in which the “HF” hydraulic fracturing pressure levels are not reached, and the volumes injected are not defined a priori by an ‘HF’ type calculation. Seismicity results from a sliding process on pre-existing fractures, sometimes called Hydro-Shearing, which has a positive irreversible effect on permeability. However, several projects (in Europe and Korea) have been halted because the induced earthquake was too great. The question is: could a study of the characteristics of the swarm of seismic events have made it possible to anticipate and stop an injection process in time?

We would like to thank the reviewer for their time and efforts taken in critiquing our work. They have provided a nice synoptic overview that sets the stage for our work, which starts addressing this unanswered question of ‘stopping in time’. As well, the reviewer has been quite thorough in their

examination of our paper and asked a series of attentive/insightful questions, so we are grateful to have this level of experience/detail for improvements.

In the responses below, we provided detailed rebuttals to each of the reviewer's suggestions that outline the changes we've made to our manuscript.

More generally, the proposed article aims to promote an important piece in a work flow for a TLS “traffic light system” that would be more reliable than the algorithms already proposed. The article is perplexing, as it could be more systematic. As it stands, the sites selected to demonstrate the capabilities seem too well chosen. Several of the co-authors have, for example, worked on Iceland, a country that is ahead of its time in the use of geothermal energy, not “HF”, and whose databases could have been (re)used (Cf Broccardo, M.,, Karvournis D., .., Wiemer, S., 2020, <https://doi.org/10.5194/nhess-20-1573-2020>).

On this critique, we would like to point out that this study shared was a pre-assessment of anticipated risk at the Geldinganes operation in Iceland. A post operation assessment was never performed because insufficient data was recorded there. Because of this, we cannot perform our analysis on this case.

More generally, the Icelandic cases don't make great candidates for this type of analysis: they are closer to conventional geothermal, targeting previously fractured/faulted rocks and sometimes even injecting under gravity-fed conditions. Furthermore, these operations are often long-lasting and mixed in with concurrent natural seismicity – making it difficult to link event clusters to a specific injection timeframe. On the other hand, for this study, we're trying to look for cases that have undergone some sort of stimulation process, that could serve as a mechanism to bound the growth of the magnitudes. In this sense, we need cases that capture all the events from the very start of injection and have cleaner links to their causal stimulation. Given those criteria, we have been exhaustive with the publicly available catalogues (to our knowledge).

On this point, we don't feel that our cases are necessarily 'well-chosen'. To explain, we'd like to provide some background context. Prior works that first developed this idea [Schultz, 2024] already applied the approach to all of the high-profile cases that approach McGarr's upper limit. These cases include Basel, Pohang, Peace River, Pawnee, Prague, Youngstown, Paradox Valley, etc. All of those cases were discerned as unbound. In retrospect, this makes sense, since these also tended to be cases where the induced earthquakes went 'poorly'.

Afterwards, it was in applying this approach to the data at FORGE that we discovered the first bound case. This gave us a hunch as to the bounding mechanism (i.e., Figure 8), which we started looking for similar cases that hadn't yet been examined. Thus, St1, PNR-1z, (and SSFS) were chosen – essentially stimulation cases where induced seismicity management 'seemed to work'. Together, all these cases created a consistent and repeatable narrative. Furthermore, we feel that this variety of operations, settings, and cases was already a series of compelling examples. Said another way, if we examined another (Icelandic) case, regardless of if the result is bound/unbound, it won't change the results/interpretation of this study. Based on this, we felt there was an important finding to report – that bound cases do exist, and they probably contribute to smoother operations.

It is always true that we could have examined more cases (as available), but there is a trade-off between explaining the fundamentals of the concepts/methods and repeating the methods for another case. Furthermore, as Reviewer #3 points out, finding bound cases is known to be a difficult problem. Thus, we wanted to focus on being crystal clear that the method does work (and is expected to work). With this paper, we feel that we have found a nice balance between these two trade-offs. In our opinion, examining more cases could be part of follow-up works.

In response to this, point we have provided a better description of the prior unbound findings (lines 320-322), to also give future readers this background context. We note that this change is similar to a comment from Reviewer #1, where they asked about applying CAP-tests to the Basel and Pohang cases.

The conclusion could be more nuanced. It is insisting on the fact that a necessary condition for de-risking an “HF” operation could be to identify as soon as possible “a unbounded growth process”, a point the « CAP » package can do, but will this be sufficient ? An example is the case of SSFS illustrated in the supplementary materials, where the likelihood of « bound growth » was established but accompanied by ruptures with magnitudes greater than 2.7. In the end, wouldn't it be better to cap the fluid volumes involved at a few thousand m³ in all cases, in the sense of Galis, for example? Or shouldn't the authors announce a combination with more quantitative approaches, once a degree of certainty about the likelihood of the situation "Mmax_bounded" is obtained?

This is a fair point raised by the reviewer. While we have tried to be clear that CAP-tests aren't the only tool that will be needed in the future, there are parts where we could be clearer. To address it, we have

revised the wording in the discussions (Section 4.1) to emphasize that, while CAP-tests are important, they will likely be part of a suite of tools that are needed to de-risk induced seismicity.

Note that we have refrained from addressing the comment regarding volume caps right here. Instead, we respond in a later comment, where the reviewer asks a similar question.

The text also leaves the impression that the ‘HF’ technique would be the main alternative for renewable geothermal energy, which is not yet demonstrated. The authors do not mention that HF technology has already been banned in some countries. Only one plant has been launched since the end of 2024, based on the experiments carried out at Utah FORGE. (FERVO Energy's Cape geothermal project, Utah) and to date there has been no feedback on the long-term validity of the system.

In the text, we’re not trying to imply that this will be the only (or even main) renewable alternative – rather that it could be an important part of that portfolio. We have revised parts of the introduction to try and be clearer on this point. If the reviewer has specific phrases/sentences that left this impression, we would be happy to discuss/revise further.

Minor comments throughout the text:

Abstract:

In the abstract line 18 , 20 and 23 , introduction , line 5, 23, etc, and caption figure 8 : the wording ‘stimulation’ is inappropriate. Hydraulic fracturing « HF » requires an injection pressure above the value of the minimum principal component of the stress tensor, to create a new fracture in tension // at SH_max ... (grey polygon in fig 8). The word “stimulation” has more to do with modifying the properties of a pre-existing fracture network, or even improving its connectivity, which is achieved at lower pressures (the “fault reactivation” case in figure 8). In figure S7, the caption is again imprecise: it would be preferable to call the Helsinki St1, Utah FORGE and SSFS “geothermal project” instead of EGS, and the UK PNR “Oil&gas”, and to reserve HF for the case where a reservoir is developed in stages using hydraulic fracturing with proppants (Utah FORGE, PNR1&2) or not (Helsinki ST1).

Similar to a comment from Reviewer #1, we have made an explicit definition of what we mean by stimulation – to more precisely include both hydraulic fracturing for tensile failures and hydro shearing. We feel that this suitably addresses the comment.

Given this distinction, we have left the EGS labels for St1, FORGE, & SSFS as is – since these are the terms/labels that these projects have given already themselves [e.g., Dorbath et al., 2009; Kwiatek et al., 2019; Moore et al., 2019]. Thus, we would prefer to be consistent with the original operator’s terminology.

1 Introduction

This section contains the technical description of the new work flow, which is based on a series of statistical tests for analysing sequences of induced events recorded over time (CAP tests). According to the Gutenberg-Richter ‘GR-MFD’ law, a maximum potential magnitude is statistically predictable, but remains a function of the number of events in the catalogue. The authors suggest a methodology for testing the likelihood of the existence of a stopping condition (a physical process) that would guarantee in advance a maximum value M_{max} for the magnitude, lower than the ‘GR’ forecast. They summarise several test cases detailed in parts 2 and 3 and show that this is possible. The best classical model prediction is even obtained, based on cumulative injection volumes (MacGarr et al., 2014 or Galis et al., 2017). On the other hand, the analysis makes it possible to detect in advance so-called “unbound” situations in which the rupture mechanism would become uncontrollable, resembling the reactivation of a fault, with a probable magnitude that would be unacceptable. Line 80: ΔM_{LRG} is somehow converging to 0 under the ‘bound’ option. This part is clear, but please check caption of figure 1.

The reviewer has provided a nice summary of the core concept of our work. We are pleased to hear that we were able to effectively communicate these concepts! We have read over the caption of Figure 1, and this appears accurate (notwithstanding the reviewer’s next suggestions). If the reviewer has any other concerns, we’d be happy to address them.

Figure 1: The ‘upper panel’ caption (left/right) does not seem to correspond to the labels (respectively bound/unbound). Also, the ΔM_{LRG} catalogue contains far fewer elements than the M catalogue. The axes on ‘middle panel’ should transcribe this. ‘Bottom panel’ confirms how a new value of ΔM_{LRG} can change the balance on the likelihood of the bound/unbound hypothesis. It

might be useful to show the point in the series at which the ‘certainty’ of the ‘bound’ hypothesis is favoured.

Thanks to the reviewer for catching this, they are correct that the bound/unbound labels were reversed in the caption of Figure 1. This is now corrected.

We now explicitly mention that ΔM_{LRG} will have fewer elements than M , in the supplementary materials (Section S1).

We have added a sentence to the caption that lets the reader know when bound/unbound hypotheses are favoured over the other.

Line 85: Weight EW-tests. There is no clear justification for applying the test to the cumulative catalogue of events from several successive sequences, or to clusters, as in the case of FORGE (Fig. S15 and S17), when the injection site is moved along the borehole. Is the statistical method still valid with small data sets ? In figure S17, please, check caption for cluster 3.

On the point of justifying cluster choices, we would like to kindly point the reviewer to the supplementary materials. There, we provide these justifications in Section S5, where each sub-section covers the cases (S5.X), where each X^{th} case has an explanation as to our clustering justification (see Sections S5.X.1).

Furthermore, we have a supplementary section dedicated to the perturbation of cluster assignment (Section 7.1). Briefly, there we show that if we drop out the last X stages this does not tend to change the confidence of a bound result; however, if we drop out the first X stages this does have a detrimental effect on our confidence of a bound result. This result also intuitively makes sense, given the interpretation (Figure 8). In the case where we drop out the first X stages, we are effectively ignoring the initial bound stages that were creating the fracture network and thus we are less able to discern these features via CAP-tests. When we dropout the last X stages, we do not remove these bound stages, and can still arrive at the correct result.

To be clearer on this point to future readers we have modified the text to more clearly indicate these supplementary resources.

For the validity of the method with small datasets, we refer to our response to Reviewer #1. There we highlighted the synthetic tests in this study (Section S4 & Figures S3-S6) as well as the resolution tests in prior works [Schultz, 2024]. Briefly reiterating, with good data, our CAP-tests usually need 10s-100 events before being confident in asserting a bound case.

2. Bounded growth

Section 2 analyses situations where the proposed work flow leads to the hypothesis of a mechanism limiting the maximum magnitude ('bound growth stage')

Synthetic scenarios (supplements S5 and S6) show how the Max magnitude can be estimated with confidence, according to the McGarr or Galis models. Two site specific data sets are used in the main body of the text, from shale gas and deep geothermal exploration respectively.

The reviewer is correct in their assessments here.

Note : Figure S8 and S11 are not colored according to stage, as stated.

*As a point of clarification, Figures S8 & S11 are coloured according to *cluster* rather than stage. Both PNR1z (Figure S8) and Helsinki St1(Figure S11) were handled as a single cluster, so all stages have the same colour.*

The Helsinki St1 geothermal project (Otaniemi project) from which the processed data originate (see figures 4, S11-S13) was ultimately abandoned for lack of productivity. The multi-stage HF strategy followed (Kwiateck et al. 2022, Kukkonen et al. 2023) to steer the development phase by continuous quantitative hydro-mechanical analysis was successful, since the “administrative” magnitude threshold set at 2 was not exceeded, with a maximum magnitude recorded of 1.9. But the industrial project remains a failure. The new methodology proposed here would prove that the “bound” hypothesis appears to be the most favoured, which an operator could have used to say that the risk was under control. However, the probable next jump ΔM_{LRG} cannot in this case be less than the distance to the red_light, of 0.1.

Do you think that the “proceed as planned” argument would have been convincing enough to continue the test, given that the probability was only significantly in favour of the Galis model after

several hundred events and a penultimate jump of magnitude ~1.5?

We would agree with the reviewer’s statements here! The data seems to suggest that the St1 operation was likely less hazardous than other operations (e.g., Basel or Pohang), we suggest this is because it was bounded. With a better understanding of CAP-tests, it could be possible to start identifying these bound cases and then change stimulation programs accordingly.

In fact, this was the intention with our discussion of Section 4.2 (lines 337-350). Operators who know they are in a bound situation have additional information that would give them some peace-of-mind to continue operations (as compared an operator still being in the unbound situation).

That said, while this is a future possibility, getting to this level of decision-making scrutiny will require additional research into the aforementioned nuances and caveats.

Question : Figure 4a) does not show the event with Mag 1.9. (Cf as stated in line 97) . Figure supplementary S12 and S13 show 2 events, eventually not falling into the 1.9 bin

Very good eye from the reviewer here! It turns out we put an improper label on the y-axis for Figure 4 – it’s off by 0.5 magnitude units. We’ve now corrected the plot, which is shown below for convenience. Thanks again for catching this oversight!

The largest event doesn’t fall in the 1.9 bin for Figure S12 because it has a value of M1.87, so it falls just short.

Lastly, b-values <1 are usually associated with fault structures, which tend to favour an unbound mode. Further on in section 3, the analysis of cluster 3, stage 7-10, on Utah FORGE in 2024 will

result in a “fault” type rupture, with a b_value of 1.8. Is this consistent?

We are aware of this association: that high b-values (up to 2.0) are usually from HF stage stimulations, while lower b-values (~1.0) are associated with fault structures [e.g., Maxwell et al., 2009].

Given the results of this study, we have a new interpretation for this phenomenon. Part of the reason why high b-values may be associated with stimulation stages at HF operations is because of the bounding process causing roll-off, making erroneous b-value estimates possible – especially for small catalogues. When an unbound fault structure is encountered, this bias no longer exists, and the b-value is (more) correctly constrained. We mention this point on lines 299-302.

Given this, we feel that our b-values in this paper are consistent.

The article would benefit from a more detailed account of this SSFS case in the main body of the text, as it is the only one that does not involve the “HF” technique stricto sensu. The SSFS project has been in operation for more than 10 years, with no significant contribution from the last very low-productivity well.

For this point, we respond similarly to another one of Reviewer #2’s critiques regarding the addition of Icelandic cases. We had chosen to focus on the other cases as they provided examples of fully bound cases for HF/EGS and then transition cases for HF/EGS. In this sense, we’re seeing a spectrum of similarities across a diversity of settings, cases, operations. Additional cases after this point are only supplementary in the sense that they further reinforce this main point. Thus, we chose to keep SSFS in the supplementary section.

Of the cases that we chose to place in the supplements, the SSFS case felt most appropriate. Of all of the cases, this dataset has the least number of events, has large magnitude discretization, and has apparent temporal gaps. Generally, SSFS appears to be the least robustly recorded of the four cases. Because of that, we opted to focus more on the others. We’ve added a statement to this effect on lines S615-S619, so future readers are also aware of our rationale.

Note that we moved this comment upwards in this response document. For ease of reading for the reviewers/editors.

Section 2 concludes with a note on the Soultz site, in supplementary material (Figure S27 to S36). These data do not reflect an ‘HF’ development protocol. Nevertheless, the algorithm identifies the ‘bound’ hypothesis for the maximum magnitude at all stages of the project's development:

Question: Regarding the choice of the 1993 extract (S27): What exactly is the test illustrated? There were several hydraulic sequences in 1993, with up to 10,000 events recorded. Here, there are only a hundred or so, even though there was a massive injection test that ended with a real opening mechanism, oriented in the direction of the major stress (Cornet et al., 2007)?

Generally speaking, for reference to this dataset we refer the reviewer to our “Data Availability” section. The source data is from the IS-EPOS platform (<https://episodesplatform.eu/>), which hosts a database of induced seismicity cases, one of which is SSFS.

The 1993 SSFS data starts in late August and ends in mid-October. Most of this stimulation occurs during September.

All of this information/data will also be freely accessible to future readers online, via the GitHub link we will share.

Question: The SSFS test, dated 2000, is the stimulation of the first deep well GPK2, followed by wells GPK3 in 2003 and GPK4 in 2004 and 2005. During each operation, larger volumes were mobilised (25,000, 37,000, 9,000 and 12,000 m³) and the maximum magnitude recorded exceeded the value of 2.5 at least once. Could you comment on the b_values shown in the figures for the SSFS test, which are all less than 1? Why are the values lower than those published (Charlety et al., 2007) for operations on the deep boreholes (S29, S31, S33, S35) at the Soultz sous Forêts site?

For this point, the Charlety et al., [2007] paper doesn't provide b-value estimates. We're assuming that the reviewer is referring to this work instead [Dorbath et al., 2009] (or maybe this one [Cuenot et al., 2008]?).

While we can't state definitively why prior studies would have different b-values (we don't access to these older catalogues), we can surmise. The b-values will be influenced by the type of magnitude scale used, the methods used to estimate magnitudes, and the size of the catalogue. Both referenced papers [Cuenot et al., 2008; Dorbath et al., 2009] used a duration magnitude, while our study uses moment magnitude.

We note that CAP-tests are invariant to linear transformations of the magnitude scale, from prior synthetic tests [Schultz, 2024] – so a conversion from M_D to M_w (etc) won't affect the bound/unbound results.

That said, there is a new paper that has homogenized the magnitudes for the SSFS case [Drif et al., 2024]. We have updated the paper to now use this catalogue instead for the CAP-tests for SSFS (i.e., Section 5.5 & Figures S27-S36). The b-values here are close to the ones in prior papers [Drif et al., 2024]. As well, updating to this catalogue has not changed the results/interpretations of our study.

We note that this critique is another reason we have chosen not to focus on the SSFS case in the main text of our manuscript.

In table 1 S1, 4 times a MacGarr model is considered (and the likelihood of a bounded behavior favored), whereas figure S32 for SSFS 2003 seems still at risk, with magnitudes of 2.7 recorded at the end of the operation.

Yes, in this case there is only sparse data. So, making a definitive bound/unbound statement is difficult. In this case it appears that bound would likely be favoured given more data, although this case hasn't quite surpassed the significance threshold yet.

Hypothetically, if this is all the information an operator had to go on, then proceeding as if it was unbound would be advisable.

3 The transition to unbound growth

This is a general comment: in all the cases presented, the option of an unbounded model is favoured up to the arrival of around ten events. Do you have any experience of a case with a M_{max} that was evaluated as probably “bound” up to a certain number of events (and therefore a certain volume injected) and then switched back to an “unbound” mode? This is certainly conceivable, since a number of major events have occurred after the injection was stopped, or even several months later (Cf site de Bales, Häring et al., 2008, Catalli ... Wiemer, 2013 doi:10.1029/2012GL054147) Could there be a bias linked to the fact that an “HF” test is necessarily limited in volume?

The short answer is no, we haven't observed this yet. Our idea is new, so we've had a limited amount of time to examine cases – effectively this idea puts us in new territory! That said, we are aware of this as a possibility. It's part of the reason why we drew our unbound interpretation (Figure 8, far right case) to hint at this possibility. While we haven't seen this example yet, we'll be on the look-out for it in future works! We note that this first comment is similar comment as the one made by Reviewer #1.

For the second question regarding a bias linked to limited volume from HF operations. We do not think this would be the case. In part, this is supported by larger magnitude events (M5) already being linked to limited volume operations. Examples include EGS in Pohang and HF in the Sichuan Basin of China. While volume can play a role in triggering more numerous events, it appears that being bound/unbound can also strongly influence the amount of hazard resulting from that sample of events. It is noteworthy to reiterate here that our approach found Pohang to behave as if it was unbound.

Question: If it were the case that the regulation only authorised a maximum volume for a stimulation, in the “EGS” sense, then would this purely statistical tool prefiguring a TLS be useful, since it seems to start every time by switching to the “unbound” mode?

Yes, even in this case, we feel that CAP-tests would still be useful! This is for two reasons.

The first is that McGarr-like and Galis-like relationships also predict M_{MAX} values at smaller volumes. With sensitive monitoring or deeper catalogues, it should be possible to infer bound cases earlier. Thus, low volume limits don't necessarily preclude the possibility of finding bound cases. Similar to a comment from Reviewer #1, we've added a line to the main text that explains this point now (lines 328-330).

The second is that bound cases provide some more nuance to a volume cap question. For example, an appropriate volume cap depends on the seismogenic index of the faults being reactivated – but it will also depend on if it is bound/unbound, and furthermore the exact specifics of how it is bound. This line of reasoning is in the same spirit as to what we've discussed in Section 4.2 (lines 337-350). We also note that our response here ties in with the “limited volume bias” question above.

In the case of ‘HF’ operations for shale treatments, illustrated by figures 6 and S22, PNR2-stage 5-7 test, carried out in 2024, it is difficult to understand what shutdown measures could have been taken before the major event that occurred 3 days after the shut-in (see line 204) for only 2,500 m³ used, since the “HF” treatment was identified by the work flow as most likely to have been

‘unbound’ very early on: Would it be appropriate to illustrate this case with an abscissa relative to volume or time and in // the evolution of b-value?

We agree with the reviewer that the specific linkages to decision-making are still difficult right now. That said, the indication of this cluster being unbound is an important piece of new information. The operators of PNR2 could have had an indication that this eastern cluster is more significantly hazardous than the initial stages from the western cluster (as well as PNR-1z). This information would have been known from the very first stage of that cluster (i.e., stage 5). With that, the operators could have chosen to skip the subsequent stages entirely, since they’re potentially linked to more hazardous fault structures.

Of course, this should be taken with a grain of salt, since all of these suggestions have the power of hindsight. Moving forwards, cases like this could be used to design an intelligent decision-making policy that incorporates bound/unbound inferences. We mention these points on lines 274-284.

4 Implications

Section 4 could address another major point. The question of whether the ‘stimulation’ operations analysed were hydraulically conclusive is not envisaged. Would a continuously controlled operation with a growth in microseismic activity classified as “bound” necessarily be conclusive from a hydraulic point of view? The case of Helsinki St1 proves that the ‘HF’ treatment did not achieve the objectives in the economic sense.

For this critique, we’re not completely clear on the point that the reviewer is trying to make. Specifically, because of the ‘hydraulically conclusive’ typo. So, some further clarifications are likely needed (and welcome) here.

If we’re understanding the question correctly, the reviewer is asking if there is a potential conceptual linkage between the hydraulic conductivity of the reservoir and bound/unbound inferences. That and using the overall economic viability of operations as a potential metric to infer something about the overall hydraulic conductivity of the reservoir.

At this point, we’re not sure if there is or should be a linkage between these two concepts. Certainly, it appears to be a robust feature of the data that hydraulically connected stages should be linked into a single cluster of events for CAP-testing (Section S7.1). If our interpretation is correct, then it would be

plausible that zones that are relatively more hydraulically conductive would also tend to be unbound – although, this wouldn't always have to be the case. Examining this linkage would be a great follow-up idea!

As a side note, the inferences that cases like St1 were bound could potentially give some indication for continuing an operation – meaning that continued St1 operations would have remained safer than an unbound variant. We mention this concept in Section 4.2 (lines 337-350).

Similarly, in the Utah FORGE project, the success of the 2022 operation (stage 3, figure 5a) remains. On the basis of the 2024 operations, stage 7-10, the last cluster 3 is identified by the new methodology as ‘unbound’, i.e. classified as at risk, and the operation could/should have been stopped earlier. The maximum magnitude recorded was 2.1. However, the subsequent hydraulic circulation and tracer tests (reported elsewhere by one of the co-authors) showed that the flows obtained over the entire length of the borehole (all stages) were unevenly distributed, and that the zone relating to cluster 3 played a major role in the final hydraulic efficiency obtained. Non-constant spacing and irregular contribution, are known conditions likely to promote early thermal breakthrough.

In this particular case, we respectfully disagree that this means the operation here should have been stopped earlier (or even at all). While the FORGE cluster 3 (stages 7-10) was unbound, it was still below their yellow-light threshold, and even further below their red-light threshold. Because of this, the FORGE operators still have the option to continue, just with the understanding that they need to tread more carefully than they did with the prior bound clusters/stages.

Hypothetically, if they were close to the red-light threshold, then the decision to continue/stop would differ depending on if that scenario is bound/unbound and exactly how far away from the red-light they are. Figure 7 (and lines 285-293 & 303-322) gives some insights as to how far away is far enough, for both scenarios.

Note Figure 8: The illustration of the “complex” situation is strange, because maintaining conditions in tension after a shear relay is not very realistic, unless the first grey polygon continues to extend at the same time. The ‘clustered’ case is also awkward. There is very little chance that the strongest magnitude will be on a grey polygon propagating in tension.

On the first point, we note that the grey polygons have the potential for both permanent dilation via proppant placement and small shear deformation. Here, we have amended the caption of Figure 8 to mention the broader definition of stimulation that includes both tensile and shear components of motion. We note that this point is similar to another (prior) comment.

On the second point, we have moved the clustered case's largest magnitude event to the fault polygon instead.

If the reviewer has any other changes they'd like to see made to this figure, we'd be happy to accommodate them!

Reviewer #3:

This paper addresses the problem of determining the maximum magnitude (M_{max}) of induced seismicity. In my review, I will focus on the statistical part of this paper.

We want to start by thanking the reviewer for bringing their expertise to the paper! Having another set of eyes that can give statistical insights will be important to better demonstrating the efficacy of our work. Thus, we are enthused to address the critiques below.

I want to start with the fact that several seismological papers assert the extreme difficulty in the estimation of M_{max} , e.g.:

Holschneider, M., Zöller, G., & Hainzl, S. (2011). Estimation of the maximum possible magnitude in the framework of a doubly truncated Gutenberg–Richter model. *Bulletin of the Seismological Society of America*, 101(4), 1649-1659.

Zöller, G., & Holschneider, M. (2016). The earthquake history in a fault zone tells us almost nothing about M_{max} . *Seismological Research Letters*, 87(1), 132-137.

Zöller, G., Holschneider, M., & Hainzl, S. (2013). The maximum earthquake magnitude in a time horizon: Theory and case studies. *Bulletin of the Seismological Society of America*, 103(2A), 860-875.

We agree with the reviewer that prior work has indicated that it is extremely difficult to assert an M_{MAX} from a catalogue. In fact, we surmise that this problem hasn't been adequately inspected for induced seismicity, largely because of the general feeling in the seismological community that this problem can't be tackled.

We are aware of (most) of those papers shared. In fact, we have cited one of them already [Holschneider et al., 2011] on lines S138-S140 and elsewhere. That paper was cited there, since it nicely delineates when its either hard or impossible to assert an M_{MAX} [Holschneider et al., 2011; Equation 15 & Figure 1]. Furthermore, this impossible/hard delineation informs our pre-screening of cases via the degree-of-truncation metric during our simple-tests (lines 85-87). Essentially, this becomes our first step in trying to find bound cases.

All of that said, the reasons for being able to say something about M_{MAX} in this study is two-fold. The first reason is that CAP-tests are generally more sensitive to M_{MAX} than traditional approaches [Schultz,

2024]. This increased sensitivity is facilitated by ΔM_{LRG} , which reframes the problem to examine a parameter that is more sensitive to M_{MAX} (as compared to M). The second is related to the cases: M_{MAX} models like McGarr and Galis have made predictions that are quite testable, even by traditional M_{MAX} approaches. These models propose that M_{MAX} changes as a function of volume, which means that very small M_{MAX} values should have truncated the GR-MFD at very small injected volumes – and similarly for moderate to large volumes/magnitudes, throughout the entire injection history. This entire process should have a serious (and quantifiable) impact on the final GR-MFD (e.g., Section S4 & Figures S3-S6). Said another way, these cases are truncated well enough to constrain M_{MAX} . In fact, we now show this effect more clearly to the reader in the newly added Figure 9.

In retrospect, we feel that we could have conveyed this difficulty to reader better. We have significantly revised the main text introduction to more explicitly mention this difficulty and cite these papers (lines 78-87) – we have added a new paragraph there, right before mentioning our novel solution via ΔM_{LRG} .

Last, we want to emphasize that this difficulty is part of what make this work important and impactful (in our opinion).

In this work, it seems that with some induced seismicity catalogs, it is possible to infer the existence of a maximum magnitude. However, to properly infer this existence, the other fundamental parameters of the Gutenberg-Richter law must be robustly estimated: the magnitude of completeness (M_c), and the b -value.

I found some important flaws regarding these Gutenberg-Richter parameters:

On this point, the reviewer is mostly/partly correct (just one small nuance is needed). The determination of b -value and magnitude-of-completeness are important metrics for the simple-tests, MLE-tests, and EW-tests. The simple-test uses traditional GR-MFD approaches, and thus need these metrics. The MLE-test and EW-test also assume a GR-MFD model, and thus need these metrics as well.

*However, the b -value is not important for our KS-test. In fact, we don't even need to know it to perform this test! Our KS-test is completely data-driven: it just compares the distribution of M versus ΔM_{LRG} , which are two catalogue observables. This means that even if we were *completely* erroneous in determining the b -value, the KS-test would still provide useful information. We more explicitly state this fact to future readers on lines S126-S127 now.*

Generally, this has guided our philosophy for building CAP-tests: amalgamate a series of fundamentally different statistical tests, each with their own unique pros/cons, which are intentionally chosen to cover deficiencies of the others. In this sense, when all the CAP/simple-tests suggest a similar bound/unbound result, we can be (more) confident that we have reached the right interpretation – even if there might be data/method issues. Conversely, conflicting interpretations likely means that we need to dig more into data/method issues.

We note that all our simple-tests and CAP-tests have corresponding results, for every one of the main text cases.

1) In the Supplementary Material, the authors state: “The M_c is simply evaluated by examining the peak bin of the non-cumulative GR-MFD and then setting conservatively.” Such a type of estimation is not proper in my opinion, given the importance of M_c in the b-value estimation and the count of the number of available events. The authors should use modern techniques, such as those based on the exponentiality of data:

Herrmann, M., & Marzocchi, W. (2021). Inconsistencies and lurking pitfalls in the magnitude–frequency distribution of high-resolution earthquake catalogs. *Seismological Society of America*, 92(2A), 909-922.

Taroni, M. (2023). Estimating the magnitude of completeness of earthquake catalogs using a simple random variable transformation. *The Seismic Record*, 3(3), 194-199.

Moreover, the authors did not investigate the short-term incompleteness problem, which also affects induced seismicity catalogs, and the corresponding b-value estimation.

On this point, we don't expect this to change the results of this study. In part because of what we've described above with the KS-test. Also, in part because we dither our magnitudes/ M_c during the bootstrap process of the simple/CAP-tests (Section S5). Also, because we have already performed a perturbation analysis on the value of M_c for every single case (Sections S5.X.2). In this perturbation analysis, if we shift the value of M_c by ± 0.2 , the results/interpretations do not change for any case. Thus, we had felt that our study's results were robust (regarding M_c).

That said, to be exhaustive on this issue, we have changed our M_c selection process. Instead, we now choose the M_c value that maximizes the goodness-of-fit to the GR-MFD. The text describing the M_c selection process has been updated to reflect this (lines S62-S70), including a relevant citation. For reference, an example of this M_c selection process is provided below.

In all cases the M_c has not changed significantly (no more than 0.1 magnitude units). This optimization process has tended to increase the confidence of our results; previously, we were being overly pessimistic/cautious with our M_c choice.

Overall, we find that using optimized M_c values does not meaningfully impact the results/interpretations of our paper. This is intuitive, as the GR-MFDs visually appear well-behaved at the low-magnitude ends, near our (new and old) M_c choices.

If/when this paper is accepted, we will also share all of the routines/codes used to perform this (and all other) analyses.

2) According the Table S1, the estimated b-values span from 0.53 to 1.88. This is a very large variability, which could be caused by an improper estimation of the Mw magnitude for such small events, or an incorrect estimation of the magnitude of completeness.

It's true that there's significant variability between these b-value estimates. That said, we would not expect these disparate cases to have similar b-values. Each case is located in a different setting, has differing operations, and varied stimulation programs. If you consider the clusters that are associated with a single case, then this variability is significantly diminished – most of the clusters have b-values similar to their case's average (Table S1). If we're going one step further, prior results from HF cases have noticed that operations on a single pad can have b-values that vary from 2.0-to-1.0 between stages [Maxwell et al., 2009]. So, this level of variability has precedent, even within a single fixed operation/setting.

More broadly speaking, it's always possible that there's errors in the input datasets (e.g., improper Mw estimations). In this study, we have been exhaustive in trying to account for these types of errors. For example, we've considered dithering our magnitudes, dithering our magnitudes-of-completeness, errors on b-values (Section S5), and have even substituted cases to use entirely different catalogues that were processed with different methods (Section S6). None of these considerations have impacted the results of the study.

Following this principle of mistrusting the data, we also repeated our CAP-tests for a variety of cases, settings, and operations. All of these cases paint a consistent picture for bound/unbound cases (Table S1). Also, in responding to a comment from Reviewer #2, we have used a new homogenized-magnitude catalogue for SSFS and found the same results as before.

Based on all of this, we feel that the chances that there is a consistent error lurking in all of the cases/datasets is slim.

3) In this work, another way to model the right tail of the Gutenberg-Richter distribution, i.e., the Tapered Gutenberg-Richter model, is completely neglected. This model is widely used in statistical seismology studies and gives some advantages with respect to Mmax in its estimation. See, e.g.:

Kagan, Y. Y. (2002). Seismic moment distribution revisited: I. Statistical results. *Geophysical Journal International*, 148(3), 520-541.

Kagan, Y. Y., & Schoenberg, F. (2001). Estimation of the upper cutoff parameter for the tapered Pareto distribution. *Journal of Applied Probability*, 38(A), 158-175.

For this point, we have added a new discussion (Section 4.4) and figure (Figure 9) to the paper. There, we briefly show the improvements to the data fit that a tapered GR-MFD can have versus a truncated GR-MFD. Note that this addition is incorporated as a simple toy model, mostly to give the reader a sense of future work/directions.

This change also addresses a suggestion raised by Reviewer #1 about seeing an example of the GR-MFD data in the main text.

To conclude, I suggest that the authors recompute all the M_c using a more suitable method, check the reliability of the magnitudes used in their calculation, and consider the Tapered Gutenberg-Richter model (at least for some comparison test).

Since these points are fundamental to understanding the reliability of their results, I suggest a major revision for this paper.

We would like to thank the reviewer for their statistical insights. Their suggestions were both fair and integral to improving our manuscript. We are looking forward to continued critiques and discussion, as this manuscript progresses!

Dear Editors of Communications Earth & Environment,

Thank you for taking the time to handle our manuscript during its revision process. We are pleased to hear that the reviewers feel our revisions to the paper were adequate. We have followed up here with the lingering critiques that remain.

*On the following pages are our itemized responses to the reviewer's comments. The original comments are in **bold-face**, our responses are in italics, and callouts to the original text are in "blue-face."*

Thank you again,

- Ryan & co-authors

Reviewer #1:

My comments have been mostly answered. I only have minor comments remaining, although I think it is important for the other reviewers to also be satisfied.

We would like to thank the reviewer for their suggestions, as well as their generally favourable outlook on this manuscript. We are glad to hear that we have (mostly) addressed their concerns :)

Regarding the comment about the discernible differences between bound and unbound cases by eye, the authors would like to emphasize the difference in distributions between the two cases. Currently they state this as (line 295), “Theoretically, unbound dMLRG values are expected to follow the same distribution as their magnitudes; on the other hand, bound dMLRG values deviate from their GR-MFD [32] – because MMAX restricts bound dMLRG values, creating apparently steeper b values (Figure 1).” I think this is quite clear. Maybe the part “as their magnitudes” could be more explicit; something like “as the earthquake magnitudes on which they are based”.

Thank you, we have modified this sentence to add the word “underlying” now.

Regarding the comment about a rolled-off frequency magnitude distribution that still manages to produce a large event (e.g., Goertz-Allmann and Wiemer, 2013 Geophysics), and looking at some examples from the supplementary material (Figures S9-10, S12-13, S17-21), as you say it does seem like bound cases do have this roll of that unbound cases do not have. For that reason, it seems likely to me that Figure 15 of Goertz-Allmann and Wiemer might cause problems for the model. This seems to be related to certain concerns of reviewer 3.

On this point we are slightly confused, sorry. That paper [Goertz-Allman & Wiemer, 2013; <https://doi.org/10.1190/geo2012-0102.1>] doesn't actually have a Figure 15; it only goes up to Figure 14. Only Figure 10 in there has a GR-MFD and these don't seem to have the effect that the reviewer is referring to. These GR-MFDs don't have strong roll-offs. Is the reviewer referring to a different Goertz-Allman work perhaps?

The paper mentioned in the previous/first round of comments [Goertz-Allman et al., 2017; <https://doi.org/10.1002/2016JB013731>] does have a strong roll-off shown in Figures 13 & 15. Although, that paper doesn't have a large event that followed these roll-offs.

Our prior response to this comment was trying to say that having a roll-off in the GR-MFD is necessary condition for bound cases, but not a sufficient condition. Said another way, there could be alternative reasons for seeing this kind of roll-off. For a hypothetical example, maybe there are two sets of faults producing earthquakes – F1 that is tectonically bound and very productive and another F2 that is effectively unbound but less productive. The aggregate catalogue would initially appear to have a roll-off (being F1 dominated) until F2 produces magnitudes that surpass the F1 limits.

CAP-tests start going into more detail than just looking for the GR-MFD roll-off. It also looks at the sequence of event magnitudes. I mention this, because CAP-tests can still potentially flag a case as unbound, even if it has a strong roll-off. For example, if the largest event occurred as the very first recorded event in the sequence (or at least very early on). This would give a strong indication that something else is at play.

A dedicated study would need to be done to answer these kinds of question, for each case. With how the events are clustered into distinct groups being an important part of that interpretation. To this effect, we have added a sentence on Lines 308-309 emphasizing this importance.

Applying CAP-tests to cases like CCS at Decatur could be an interesting follow-up work!

Line 349 it seems like this new phrase is not a complete sentence.

We appreciate the reviewer catching this. We have heavily revised the sentence to be clearer that we want to talk about hypothetical bound-to-unbound scenarios. That this is something future researchers should be looking for.

Concerning Figure 7, I suppose looking at dMLRG means that there are much fewer data points than when one makes a traditional frequency magnitude distribution. How many dMLRG points are needed to provide confidence in the b value they yield? Maybe this also ties in to the previous comment about bound cases becoming unbound if they run into a fault (although I see the new text regarding this case).

The reviewer is correct here that there are fewer dMlrg samples than there are M samples. The general rule for the expected number of dMlrg from an unbound M is $Nlrg = \ln(Nm) + 0.577$ [Schultz, 2024].

That said, the same general rules apply for estimating b -values from M as it does from dM_{lrg} . In the paper we have reported our b -value estimates, their goodness-of-fit, and their bootstrap error bars using traditional M -based approaches.

While we have fairly small dM_{lrg} datasets, we do have enough to be statistically confident that these two dM_{lrg} distributions in Figure 7 are distinct. Especially given that these two have been drawn from the same underlying distribution of M .

Reviewer #3:

In this revised version of the paper, the quality of the work is increased, and the authors properly replied to most of my comments. Thus, I think that this paper is now ready for publication.

Thanks again to the reviewer for their thoughts and insights!